# SAFE IN-CONTEXT REINFORCEMENT LEARNING

## ABSTRACT

In-context reinforcement learning (ICRL) is an emerging RL paradigm where the agent, after some pretraining procedure, is able to adapt to out-of-distribution test tasks without any parameter updates. The agent achieves this by continually expanding the input (i.e., the context) to its policy neural networks. For example, the input could be all the history experience that the agent has access to until the current time step. The agent's performance improves as the input grows, without any parameter updates. In this work, we propose the first method that promotes the safety of ICRL's adaptation process in the framework of constrained Markov Decision Processes. In other words, during the parameter-update-free adaptation process, the agent not only maximizes the reward but also minimizes an additional cost function. We also demonstrate[1] that our agent actively reacts to the threshold (i.e., budget) of the cost tolerance. With a higher cost budget, the agent behaves more aggressively, and with a lower cost budget, the agent behaves more conservatively.

## 1 INTRODUCTION

Reinforcement learning (RL, Sutton & Barto (2018)) is a framework for solving sequential decision making problems via trial and error. Modern RL agents are usually parameterized by deep neural networks, and the trial-and-error process is typically achieved by updating the parameters of the neural networks (Mnih et al., 2015). Recently, in-context reinforcement learning (ICRL, Moeini et al. (2025)) emerges as an RL paradigm that achieves this trial-and-error process without any neural network parameter updates (Duan et al., 2016; Mishra et al., 2018; Xu et al., 2022; Kirsch et al., 2023; Lin et al., 2023; Lu et al., 2023; Raparthy et al., 2023; Sinii et al., 2023; Grigsby et al., 2024a;c; Cook et al., 2024; Dai et al., 2024; Dong et al., 2024; Elawady et al., 2024; Huang et al., 2024; Krishnamurthy et al., 2024; Laskin et al., 2023; Lee et al., 2024; Shi et al., 2024; Wang et al., 2024; Zisman et al., 2023; 2024; Song et al., 2025; Wang et al., 2025b) Specifically, the pretrained agent neural network takes as input not only the observation at the current time step but also an additional context. For example, the simplest choice of the context can be all the history of experiences that the agent has access to up to the current time step (Laskin et al., 2023). It is then observed that the agent's performance improves as the length of the context grows, even if the neural network parameters are kept fixed and even if the agent is evaluated on a task that it has never encountered during the pretraining process (Laskin et al., 2023). This phenomenon is called ICRL. A widely accepted hypothesis for this performance improvement is that the forward pass of the pretrained agent network implicitly implements some RL algorithm during inference time to process the information in the context (Laskin et al., 2023; Lin et al., 2023; Wang et al., 2025a;b). As the context length grows, the amount of information grows, and the implemented RL algorithm in the forward pass is then able to output better actions.

ICRL has recently received increasing attention given its cheap yet strong adaptation capability. The adaptation is cheap in that it does not require parameter tuning. So to support such adaptation, the infrastructure only needs to support the inference of the neural network. For example, by using ICRL in LLMs' inference time, the LLMs are able to significantly improve on challenging tasks such as ScienceWorld (Wang et al., 2022), HMMT (Harvard–MIT Mathematics Tournament), and AIME (Mathematical Association of America) without any fine-tuning (Song et al., 2025). The adaptation is strong in that ICRL allows adaptation to not only unseen but also out-of-distribution tasks. For example, Laskin et al. (2023) show that after pretraining on some bandit tasks, their ICRL

---

[1]All the implementations will be made publicly available and are now located in the supplementary materials.

agent is able to adapt quickly to test bandit tasks that have opposite optimal arms to the pretraining bandit tasks. Kirsch et al. (2023) show that after pretraining on Ant robot manipulation tasks, their ICRL agent is able to adapt to pole balancing tasks.

Despite the empirical success and the promising potential, the safety of ICRL has long been ignored. Here we discuss safety in the framework of constrained Markov Decision Processes (Altman, 2021), where each action the agent takes not only incurs a reward but also incurs a cost. Then when an agent adapts to a new task, it needs to not only maximize the rewards but also minimize the cost, all without updating the parameters. We argue that safe ICRL is a necessary condition to enable many possible real-world applications of ICRL, such as embodied AI. Although such safe decision-making problems have been well studied in the classical RL setup (García & Fernández, 2015), to our knowledge, no prior work has studied the safety of ICRL, which is the gap that this paper shall close. We envision that a safe ICRL agent should have the following two desiderata, both of which should be achieved without any parameter updates during the adaptation to new tasks.

- When evaluated in out-of-distribution tasks, the agent should be able to maximize the rewards while keeping the cost below a user-specified threshold.
- The agent should actively react to the cost threshold. For example, a smaller cost threshold should induce a more conservative behavior, while a larger cost threshold should induce a more aggressive behavior.

The key contribution of this work is to design the first safe ICRL agent that simultaneously fulfills the two desiderata. Specifically, we make the following two contributions.

- We study both off-line (cf. (Laskin et al., 2023)) and on-line pretraining approaches (cf. (Duan et al., 2016; Wang et al., 2016; Grigsby et al., 2024a;c)) for safe ICRL. We identify key limitations of the offline approach and propose a novel online pretraining approach that fulfills the two desiderata.
- We empirically validate our proposed approach in challenging safe decision making problems. Particularly, our test task is not only unseen during the pretraining stage but also out of the pretraining task distribution. As a result, to succeed in our benchmarks, the agent has to demonstrate strong out-of-distribution generalization. Particularly, the agent has to explore efficiently in out-of-distribution test tasks, all without parameter updates.

## 2 BACKGROUND

We consider a finite horizon Markov Decision Process (MDP, Bellman (1957); Puterman (2014)) with a state space $\mathcal{S}$, an action space $\mathcal{A}$, a reward function $r : \mathcal{S} \times \mathcal{A} \to \mathbb{R}$, a transition function $p : \mathcal{S} \times \mathcal{S} \times \mathcal{A} \to [0, 1]$, an initial distribution $p_0 : \mathcal{S} \to [0, 1)$, and a horizon length $T$. At time step 0, an initial state $S_0$ is sampled from $p_0$. At time step $t$, an agent at a state $S_t$ selects an action $A_t$ according to its policy $\pi : \mathcal{A} \times \mathcal{S} \to [0, 1]$, i.e., $A_t \sim \pi(\cdot|S_t)$. The agent then proceeeds to a successor state $S_{t+1} \sim p(\cdot|S_t, A_t)$ and obtains a reward $R_{t+1} \doteq r(S_t, A_t)$. This process continues until the time $T - 1$, where the agent executes the action $A_{T-1}$ and receives the last reward $R_T$. We use $\tau \doteq (S_0, A_0, R_1, S_1, \ldots, S_{T-1}, A_{T-1}, R_T)$ to denote an *episode*. We use $k$ to index episodes and $t$ to index time steps within an episode. When necessary, we write $S_t^k$ to denote the state at time step $t$ within the $k$-th episode. $A_t^k$ and $R_t^k$ are similarly defined. For any episode $\tau$, we use $G(\tau) \doteq \sum_{t=1}^{T} R_t$ to denote the return associated with this episode. The performance of a policy $\pi$ is then measured by the expected total rewards $J(\pi) \doteq \mathbb{E}_{\tau \sim \pi}[G(\tau)]$. One fundamental task in RL, called control, is to adapt the policy $\pi$ to maximize $J(\pi)$.

Modern RL methods solve the control problem via parameterizing the policy $\pi$ with neural networks (Mnih et al., 2015). We denote such parameterized policy as $\pi_\theta$, where $\theta$ denotes the parameters of the neural network. Typically, $\theta$ is updated at every time step, e.g., $\theta_{t+1}$ is obtained by updating $\theta_t$ with the newly obtained information $(S_t, A_t, S_{t+1}, R_{t+1})$. The performance of the policy $\pi_{\theta_t}$ improves along with such parameter updates. In other words, the reinforcement learning process occurs with the parameter updates. ICRL is an emerging paradigm that allows the reinforcement learning process to occur without any parameter updates. In ICRL, the policy $\pi$ takes an additional context as input. We denote the context at time $t$ within the $k$-th episode as $H_t^k$ and express the policy as $\pi(\cdot|S_t^k, H_t^k)$ to emphasize such dependency. The construction of the context is an active research area and we refer the reader to Moeini et al. (2025) for a detailed survey. The simplest example is to

use all the history as context, e.g., $H_t^k \doteq (\tau_1, \ldots, \tau_{k-1}, S_0^k, A_0^k, R_1^k, S_1^k, \ldots S_{t-1}^k, A_{t-1}^k, R_t^k)$. After some pretraining procedure on multiple MDPs to be detailed soon, we can obtain a parameter $\theta_*$. It is observed that the performance of $\pi_{\theta_*}(\cdot|S_t^k, H_t^k)$ improves as the context grows with $\theta_*$ kept fixed. In other words, the reinforcement learning process now occurs without any parameter updates. This cannot be attributed to the hypothesis that $\theta_*$ memorizes a good policy, as such performance improvement is also observed even when $\pi_{\theta_*}$ is evaluated in out-of-distribution MDPs that are very different from the pretraining MDPs (Laskin et al., 2023; Kirsch et al., 2023). A widely accepted hypothesis is that the neural network $\theta_*$ implicitly implements some RL algorithm in its forward pass, such that it can process the context with the RL algorithm in the inference time (Moeini et al., 2025). As the context grows, the inference-time RL algorithm in the forward pass gains access to more information. So the performance of the policy improves. This hypothesis is also theoretically backed by Lin et al. (2023); Wang et al. (2025a;b).

We now elaborate on the pretraining methods for ICRL. According to the taxonomy in Moeini et al. (2025), the pretraining can be divided into supervsied pretraining (Laskin et al., 2023; Lin et al., 2023) and reinforcement pretraining (Duan et al., 2016; Wang et al., 2016; Grigsby et al., 2024a;c; Wang et al., 2025a;b). Supervised pretraining is essentially behavior cloning but the goal is to imitate an algorithm instead of a policy. Supervised pretraining is usually done in an offline manner. Specifically, by running an existing RL algorithm on an MDP, we can collect a sequence of episodes, denoted as $\Xi \doteq (\tau_1, \tau_2, \ldots, \tau_K)$. We call $\Xi$ a *trajectory*. Suppose the RL algorithm behaves well, we would expect that the episode return $G(\tau_k)$ increases as $k$ increases. So the trajectory $\Xi$ demonstrates the learning process of the RL algorithm in the MDP. In supervised pretraining, we collect multiple trajectories by running multiple existing RL algorithms on multiple MDPs, yielding a dataset $\{\Xi_i\}$. The policy $\pi_\theta$ is then trained with an imitation learning loss. In other words, for a state $S_t^k$ from a trajectory $\Xi_i$ in the dataset, the loss for updating $\theta$ is

$$-\log \pi_\theta(A_t^k|S_t^k, H_t^k). \tag{1}$$

By asking the neural network to imitate the behavior of RL algorithms demonstrated in the dataset, we expect the neural network to be able to distill some RL algorithm into its forward pass. This is known as algorithm distillation (Laskin et al., 2023). Instead of distilling existing RL algorithms, reinforcement pretraining asks the network to discover its own RL algorithm. Reinforcement pretraining is typically done in an online manner. At time step $t$ within the $k$-th episode, the loss is

$$Loss_{RL}(\pi_\theta(\cdot|S_t^k, H_t^k)), \tag{2}$$

where $Loss_{RL}$ can be any standard RL loss for standard online RL algorithms, e.g., Grigsby et al. (2024a) use a variant of DDPG (Lillicrap et al., 2016), Elawady et al. (2024) use a variant of PPO (Schulman et al., 2017), and Cook et al. (2024) use Muesli (Hessel et al., 2021). Since the context is typically a long sequence, Transformers (Vaswani et al., 2017) or state space models (Gu & Dao, 2024) are usually used to parameterize the policy (Laskin et al., 2023; Lu et al., 2023). The long sequence context is one of the main driving forces for the remarkable generalization capability of ICRL.

## 3 SAFE IN-CONTEXT REINFORCEMENT LEARNING

While ICRL has demonstrated remarkable generalization in out-of-distribution tasks, the safety of such generalization has been largely overlooked. This paper is the first to study the safety of ICRL. Particularly, we use the constrained MDP (CMDP) framework (Altman, 2021). In addition to the reward function $r$, we also have a cost function $c : \mathcal{S} \times \mathcal{A} \to \mathbb{R}$. At a state $S_t$, the agent executes an action $A_t$ and receives both a reward $R_{t+1}$ and a cost $C_{t+1} \doteq c(S_t, A_t)$. An episode is then

$$\tau = (S_0, A_0, R_1, C_1, S_1, \ldots, S_{T-1}, A_{T-1}, R_T, C_T). \tag{3}$$

We use $G_c(\tau) \doteq \sum_{t=1}^T C_t$ to denote the toal cost in the episode $\tau$. After some pretraining procedure, we obtain a policy $\pi_{\theta_*}$. We then execute this policy in a test MDP for multiple episodes $\tau_1, \ldots, \tau_K$. With ICRL, we can expect that the episode return $G(\tau_k)$ increaes as $k$ increaes. But for safe ICRL, we would like to additionally see that the episode total cost $G_c(\tau_k)$ decreases as $k$ increases, ideally below some user-given threshold, say $\delta$. Moreover, when the user is more tolerant of the cost (i.e., with a larger $\delta$), the agent should obtain higher rewards (i.e., larger $G(\tau_k)$). We formalize these desiderata as the following constrained problem:

$$\max_\theta \mathbb{E}_{\pi_\theta}\left[\sum_{k=1}^K G(\tau_k)\right] \quad \text{s.t.} \quad \forall k, \mathbb{E}_{\pi_\theta}[G_c(\tau_k)] \leq \delta. \tag{4}$$

Here, $K$ is the number of episodes that are allowed in the test time. $\tau_k$ is the $k$-th epsiode sampled from the policy $\pi_\theta$. The policy $\pi_\theta$ can depends on the context (i.e., $\pi_\theta(A_t^k|S_t^k, H_t^k)$) but the neural network parameters $\theta$ is fixed for all the $K$ episodes. We note that (4) is defined on a single test MDP that the algorithm has no access to during pretraining. This test MDP is usually sufficiently different from the MDPs that the algorithm has access to during pretraining. As a result, the agent cannot solve this test MDP via memorizing some good policies in the pretraining MDPs. Instead, it has to perform reinforcement learning over the $K$ episodes, although the parameters $\theta$ are fixed. Notably, although we formulate our problem in MDPs, the tasks we work on are only partially observable. This is a widely used practice in the RL community to promote the clarity of presentation (Mnih et al., 2015). We now explore different pretraining methods for solving the above problem.

**Safe Supervised Pretraining.** We first establish safe supervised pretraining as a baseline approach. Similar to naive supervised pretraining, we collect a dataset $\{\Xi_i\}$ by executing various safe RL algorithms (e.g., Tessler et al. (2018); Ray et al. (2019)) in various safety-sensitive tasks (e.g., Ji et al. (2023); Gu et al. (2025)). Now each trajectory $\Xi_i$ consists of multiple episodes and each episode contains both the reward and the cost (cf. (3)). To further guide the action generation, we additionally condition the policy on both the return-to-go (RTG) $G_t(\tau)$ and the cost-to-go (CTG) $G_{c,t}(\tau)$. RTG is the total future rewards in the current episode, i.e., $G_t(\tau) \doteq \sum_{i=t+1}^{T} R_t$, and CTG is the total future cost in the current episode, i.e., $G_{c,t}(\tau) \doteq \sum_{i=t+1}^{T} C_t$. The RTG and CTG are inspired by Liu et al. (2023) for constrained decision Transformers and are designed to bias the agent towards achieving the specified return and cost. The main difference from Liu et al. (2023) is that in addition to RTG and CTG, we use entire episodes in input, while Liu et al. (2023) only includes states and actions but do not include per-step reward and per-step cost. As a result, Liu et al. (2023) still misses the opportunity to learn the algorithm demonstrated in the dataset, and they demonstrated only limited generalization capability. Specifically, given a trajectory $\Xi = (\tau_1, \ldots, \tau_K)$ in the dataset, the safe supervised pretraining loss at a state $S_t^k$ is

$$-\log \pi_\theta(A_t^k|S_t^k, H_t^k, G_t(\tau_k), G_{c,t}(\tau_k)). \tag{5}$$

We recall that the context $H_t^k$ is defined as $(\tau_1, \ldots, \tau_{k-1}, S_0^k, A_0^k, R_1^k, C_1^k, \ldots, S_{t-1}^k, A_{t-1}^k, R_t^k, C_t^k)$. In the pretraining, we do have access to RTG and CTG since the episode is already complete. But in the test time, we do not have access to them since they are calculated based on future rewards and costs, which are not available yet. Instead, we replace them with a spectrum of target RTG and CTG values. This gives the user the opportunity to control the trade-off between rewards and costs in the test time without fine-tuning the parameters. For example, in the test time, if the ratio RTG / CTG is small, it means the user is more risk-tolerant. If the ratio RTG / CTG is large (in the extreme case, CTG = 0), it means the user is more safety-focused.

Although the proposed safe supervised pretraining approach is to our knowledge novel, it is indeed a straightforward extension of existing unconstrained superivsed pretraining approaches like Laskin et al. (2023). We regard this as a baseline approach for sanity check. Any meaningful safe ICRL method should at least outperform this baseline.

**Safe Reinforcement Pretraining.** Our safe supervised learning follows the offline pretraining paradigm. It is well-known that the performance of offline trained policies is usually worse than the online trained ones when online training is plausible (Vinyals et al., 2019; Mathieu et al., 2021). Motivated by this, we now study safe reinforcement pretraining. We note that even when we consider online pretraining, we still do not have access to the test MDP where (4) is defined. By online pretraining, we mean that we can interact with the pretraining MDPs directly. By contrast, offline supervised pretraining has access to only datasets collected from pretraining MDPs.

The underlying hypothesis of reinforcement pretraining is that by minimizing the loss (2) in a wide range of pretraining MDPs, the sequence model should be able to discover its own RL algorithm and implement it in the forward pass (see Lu et al. (2023); Grigsby et al. (2024a;c) for empirical evidence and Wang et al. (2025b) for theoretical evidence). Following this hypothesis, we now design an algorithm to solve (4) on a single pretraining MDP. To solve (4), primal-dual approaches in safe-RL (Achiam et al., 2017; Tessler et al., 2018; Liang et al., 2018; Ray et al., 2019) convert it to the dual

problem below[2]:

$$\min_{\boldsymbol{\lambda} \succeq 0} \max_{\pi} L(\pi, \boldsymbol{\lambda}) = \min_{\boldsymbol{\lambda} \succeq 0} \max_{\pi} \left[ \mathbb{E}_\pi \left[ \sum_{k=1}^{K} G(\tau_k) \right] - \sum_{k=1}^{K} \lambda_k (\mathbb{E}_\pi [G_c(\tau_k)] - \delta) \right] \quad (6)$$

where $\boldsymbol{\lambda} \succeq 0$ is a componentwise inequality for the Lagrangian multipliers. Notably, the function class of the policy $\pi$ in $\max_\pi$ is all the policies in the form of (5) but without RTG, i.e., the policy can depend on the context and the CTG. However, we observe that while the CMDP formulation requires a separate constraint and Lagrangian multiplier for each evaluation episode $\tau_k$, this is not desirable in our problem, as it would need to fix the number of evaluation episodes at the beginning of pretraining. During pretraining, if we expect to evaluate over $K$ episodes, we should initialize with $K$ (or more) multipliers. If pretraining rollouts contain fewer episodes than the number of multipliers, the later multipliers are updated on a slower timescale, which destabilizes the optimization (Figure 5.(c)).

Furthermore, there should be symmetry among all the constraints since they are all coming from a single cost function but simply correspond to different episodes. These encourage the use of a single multiplier for all episodes. However, when using a single multiplier, we would need a new way to not over-penalize the policy in the episodes that are already satisfying the constraint. To this end, We propose a modified Lagrangian function and an iterative optimization scheme and show it is more stable empirically while satisfying the desirable optimization properties. Let $g_k(\pi) := \mathbb{E}_\pi[G_c(\tau_k)] - \delta$. We consider the following surrogate objective for the policy

$$L_\Sigma(\pi, \lambda) = \mathbb{E}_\pi \left[ \sum_{k=1}^{K} G(\tau_k) \right] - \lambda \sum_{k=1}^{K} [g_k(\pi)]_+$$

where $[x]_+ \doteq \max\{x, 0\}$ and perform the following iterative updates:

$$\pi_{t+1} \in \operatorname{argmax}_\pi L_\Sigma(\pi, \lambda_t) \quad \lambda_{t+1} = [\lambda_t + \eta \max_k g_k(\pi_{t+1})]_+ \quad \eta > 0. \quad (7)$$

**Assumption 1.** *The expected return $\mathbb{E}_\pi \left[ \sum_{k=1}^{K} G(\tau_k) \right]$ and expected cost $\mathbb{E}_\pi[G_c(\tau_k)]$ are bounded and continuous in $\pi$. The constrained problem (4) admits an optimal feasible policy $\pi^\star$.*

**Assumption 2.** *[Paternain (2018)] Problem (4) satisfies conditions ensuring zero duality gap with (6) and there exists optimal Lagrange multipliers $\boldsymbol{\lambda}^\star \succeq 0$ such that*

$$\max_{\substack{\pi \\ s.t. \ \forall k, \mathbb{E}_\pi[G_c(\tau_k)] \le \delta}} \mathbb{E}_\pi \left[ \sum_{k=1}^{K} G(\tau_k) \right] = \min_{\boldsymbol{\lambda} \succeq 0} \max_\pi L(\pi, \boldsymbol{\lambda})$$

We now show that the set of the fixed points of (7) and the set of the optimizers of (4) are equal.

**Theorem 1.** *[Proof in Appendix B.1] We say a pair $(\bar{\pi}, \bar{\lambda})$ is a fixed point of (7) if $\bar{\pi} \in \operatorname{arg max}_\pi L_\Sigma(\pi, \bar{\lambda})$, and for all sufficiently small $\eta > 0$, $\bar{\lambda} = [\bar{\lambda} + \eta \max_k g_k(\bar{\pi})]_+$. Let Assumptions 1 - 2 hold. Then every primal-optimal policy $\pi^*$ admits a corresponding fixed point $(\pi^*, \bar{\lambda})$ for any multiplier $\bar{\lambda} \ge \|\boldsymbol{\lambda}^\star\|_\infty$, where $\boldsymbol{\lambda}^\star$ is an optimal dual solution. Conversely, every fixed point $(\bar{\pi}, \bar{\lambda})$ of (7) is feasible and primal-optimal for (4).*

To implement the policy update (7), we consider an actor-critic approach following Grigsby et al. (2024a). The pseudocode is provided in Algorithm 1. Since our update (7) resembles an exact penalty convex optimization technique when $\lambda \ge \|\boldsymbol{\lambda}^\star\|_\infty$ (Nocedal & Wright, 2006), we call the resulting actor-critic algorithm Exact Penalty Policy Optimization (EPPO).

## 4 EXPERIMENTS

We now empirically investigate the proposed safe ICRL algorithms. Specifically, we aim to answer the following two questions.

(i) Do the proposed safe ICRL algorithms demonstrate generalization to out-of-distribution (OOD) tasks in complex environments with safety constraints?

(ii) How effectively do the proposed safe in-context RL algorithms achieve flexible reward-cost tradeoffs under varying cost tolerance?

---

[2]We omit $\theta$ and use $\max_\pi f(\pi)$ instead of $\max_\theta f(\pi_\theta)$ for simplicity.

We evaluate our approach on two constrained environments: (1) SafeDarkRoom and (2) SafeDark-Mujoco: Point and Car. These are modified from DarkRoom (Laskin et al., 2023) and SafetyGym benchmark (Ji et al., 2023) respectively. SafeDarkRoom is a grid-world setting where the agent can only perceive its own position and **lacks visibility of the goal or obstacle locations**. Rewards are sparse, with a positive reward obtained upon reaching the single goal in the map. Costs arise from multiple obstacles scattered across the environment. For instance, 25 obstacles in a 9×9 grid map, which incur a cost of 1 each time the agent steps on one of the obstacles. To succeed, the agent must learn obstacle positions based on encountered cost signals, introducing additional complexity compared to goal-only environments due to the presence of multiple hazards. SafeDarkMujoco environment operates in continuous space with MuJoCo simulation (Todorov et al., 2012) but with a setup analogous to SafeDarkRoom. The robot senses its internal physical states, such as velocity, acceleration, position, and rotation angle, while **all lidar inputs are turned off, preventing detection of obstacles and the goal**. Consequently, the agent must develop exploratory behaviors to identify obstacles and goals, akin to SafeDarkRoom design, while navigating to the goal and minimizing collisions by learning from the previous collisions in the context.

**Measuring OOD generalization.** Goal discovery-oriented environments such as DarkRoom are widely used in previous ICRL works (Laskin et al., 2023; Zisman et al., 2023; Son et al., 2025). In those works, the goals are randomly spawned over the map. Each goal corresponds to a new task. By ensuring the goals used in pretraining do not overlap with the goals used in testing, they ensure the test task is unseen during pretraining and can thus evaluate the generalization capability of their ICRL agents. However, such generalization can be achieved by interpolation (Kirk et al., 2021). For example, if the goals are spawned only on the black squares of a chessboard during pretraining, then an unseen goal location on one of the white squares can be viewed as an interpolation of the goals in the pretraining tasks. The agent thus may navigate to this unseen test goal by interpolating policies learned from pretraining. In other words, although those evaluation tasks are *unseen* during pretraining, it is not clear whether those tasks can fully demonstrate the challenges of OOD generalization.

To measure the OOD generalization of our proposed safe ICRL methods, we employ a challenging distance-based obstacle and goal spawning strategy: center-oriented for pretraining and edge-oriented for evaluation. The agent spawns at the map center, with obstacles and the goal distributed proportionally closer to this center during pretraining. During evaluation, obstacles and the goal follow an edge-oriented distribution. This approach applies similarly to the SafeDarkRoom and SafeNavigation environments. We argue that this setup is more challenging than those in prior studies (Laskin et al., 2023; Zisman et al., 2023; Son et al., 2025), as the agent must learn extrapolation rather than interpolation. More precisely, during pretraining, the grid position $(i, j)$ has an obstacle with probability $\mathbb{P}_{\text{train}}((i, j)) \propto e^{-\alpha d((i,j),c)}$, where $c$ is the map center, $d(\cdot, \cdot)$ denotes Euclidean distance, and $\alpha > 0$ to promote central density. To generate an evaluation task, the grid position $(i, j)$ has an obstacle with probability $\mathbb{P}_{\text{test}}((i, j)) \propto e^{\alpha d((i,j),c)}$ with $\alpha > 0$ to favor edge density. The goals are generated similarly. This distributional shift is demonstrably OOD, both visually (Appendix C) and mathematically.

**Proposition 1** (Proof in Appendix B.2). *The total variation distance satisfies* $\lim_{\alpha \to \infty} \delta(\mathbb{P}_{\text{train}}, \mathbb{P}_{\text{test}}) = \lim_{\alpha \to \infty} \frac{1}{2} \sum_O |\mathbb{P}_{\text{train}}(O) - \mathbb{P}_{\text{test}}(O)| = 1$ *(maximum separation).* *Similarly, the KL divergence holds* $\lim_{\alpha \to \infty} D_{\text{KL}}(\mathbb{P}_{\text{train}} \| \mathbb{P}_{\text{test}}) = \infty$.

**Question (i).** We now demonstrate emergent safe learning behaviors in OOD test tasks, thus giving an affirmative answer to Question (i). We use supervised pretraining and reinforcement pretraining in Section 3 on pretraining tasks with center-oriented obstacles and goals. Particularly, for supervised pretraining, the source algorithm to generate the dataset is PPO-Lagragian (Ray et al., 2019). For reinforcement pretraining, we update the model for 30,000 steps on SafeDarkRoom and 10,000 steps on SafeDarkMujoco using Algorithm 1.

After pretraining, we evaluate the learned agents in evaluation tasks with edge-oriented obstacles and goals. The agent is able to interact with the evaluation task for multiple episodes, but cannot update its network parameters. During evaluation, for supervised pretraining, we set RTG to 1.0 and CTG to 0.0, targeting successful performance while minimizing cost violations. Reinforcement pretraining is solely conditioned on CTG. To evaluate robustness to CTG values during evaluation, we set CTG to a value uniformly sampled from the interval $[1, 15]$ for SafeDarkRoom and $[10, 50]$ for SafeDarkMujoco. Precisely, we sample 100 distinct CTG values and report the average performance

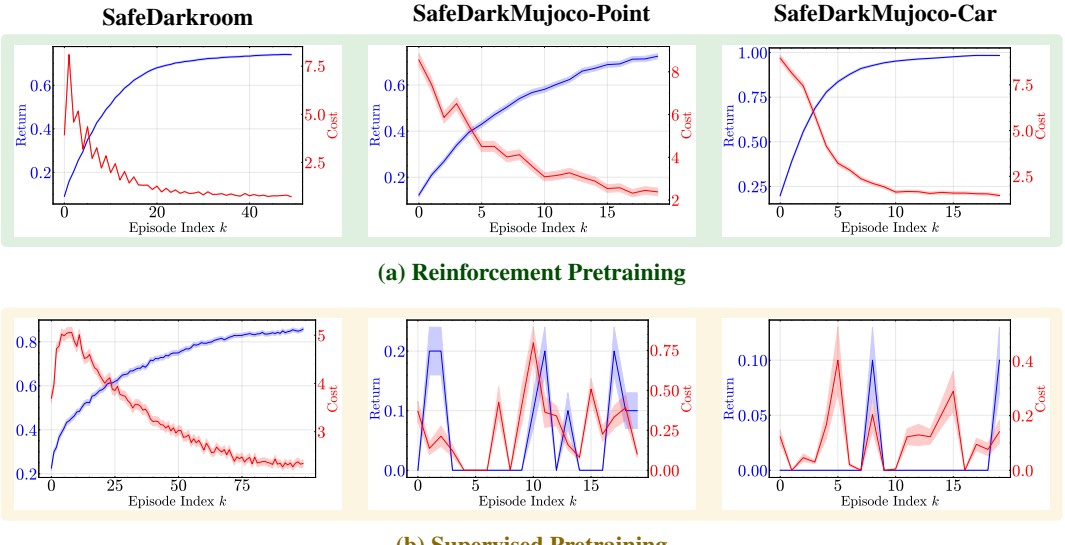

**(a) Reinforcement Pretraining**

**(b) Supervised Pretraining**

Figure 1: **Evaluation performance of superivsed and reinforcement pretraining.** The curves are averaged over 16 distinct OOD evaluation tasks, with each task featuring edge-oriented goal and obstacles. The shaded regions indicate standard errors. The $x$-axis is the episode index $k$ during the evaluation task. The $y$-axis is the corresponding episode return $G(\tau_k)$ and the episode cost $G_c(\tau_k)$ respectively. The straightforward safe supervised pretraining baseline succeeds only in SafeDarkRoom, while our novel safe reinforcement pretraining method succeeds in all three domains.

over these 100 distinct CTG values. We recall that although the safe supervised pretraining method we propose is novel, we mostly regard it as a straightforward baseline. As can be seen in Figure 1, safe supervised pretraining succeeds in SafeDarkRoom as the episode return grows and the episode cost decreases. This aligns with the previously observed success of algorithm distillation in DarkRoom (Laskin et al., 2023; Zisman et al., 2023). But this baseline approach fails in SafeDarkMujoco. We conjecture that this is because of insufficiency in the dataset, as supported by ablation studies (Figure 6). By contrast, our safe reinforcement pretraining approach consistently succeeds in all three tested domains. It is also worth highlighting that in SafeDarkRoom, the safe reinforcement pretraining enables much faster adaptation to the new evaluation task than the safe supervised pretraining approach.

**Question (ii).** We now demonstrate that we can adjust the behavior of the pretrained policy in test time by simply changing the RTG and/or CTG without making any parameter updates. Intuitively, with a higher CTG (i.e., a higher cost tolerance), the agent should afford bolder exploration and thus obtain higher rewards. Conversely, a lower CTG should promote conservative policies that prioritize safety, potentially at the expense of suboptimal rewards.

For the supervised pretrained model, we initialize RTG at 0.5 and CTG at 0.0, then adjust RTG = $\max\{0.5, \frac{CTG}{10}\}$ as CTG increases from 5 to 15 for SafeDarkRoom and 0 to 10 for SafeDarkMujoco. This setup enables gradual increases in both RTG and CTG to demonstrate balanced reward-cost trade-offs, consistently applied to SafeDarkRoom and SafeDarkMujoco. However, the supervised pretrained model fails to learn trade-off behaviors, exhibiting random patterns with varying RTG and CTG (Figure 2.(b)). We hypothesize that this stems from insufficient CTG diversity during training, leading to dataset inadequacy.

In our reinforcement pretrained model, we focus on controlling the CTG exclusively. This design simplifies the regulation of RTG based on the CTG. Simultaneously controlling both RTG and CTG is challenging, as determining the feasibility of RTG/CTG pairs is complex (Liu et al., 2023). Instead, the reinforcement pretrained model is designed to control CTG and pursue the maximal possible return for a given CTG, eliminating the need to justify the feasibility of specific solutions. Figure 2.(a) illustrates this relationship, showing that as cost limits increase, the return also increases.

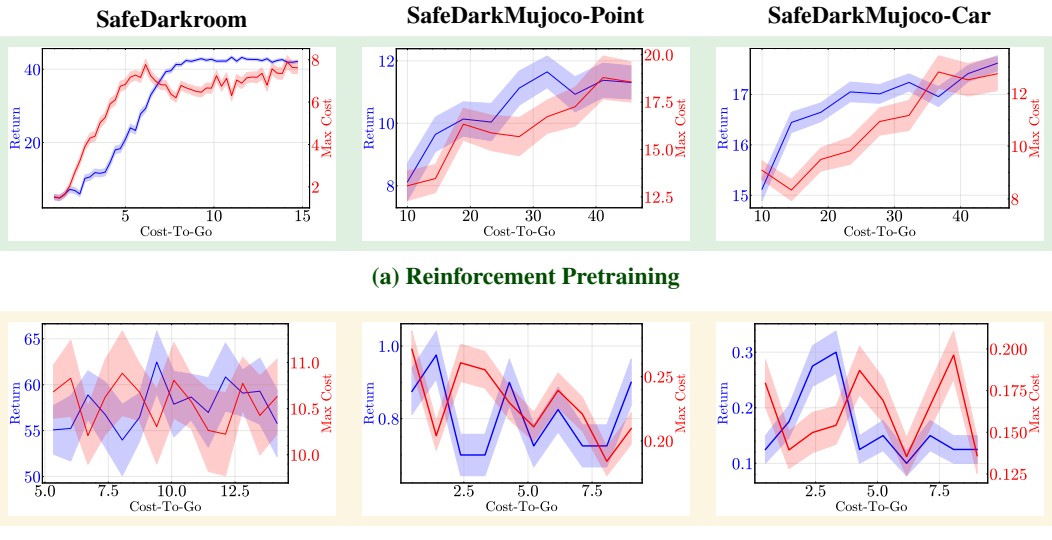

**(a) Reinforcement Pretraining**

**(b) Supervised Pretraining**

Figure 2: **Evaluation performance of supervised and reinforcement pretraining with varying CTG.** The curves are displayed across a range of cost limits. For each cost limit, the result is averaged over 16 distinct OOD evaluation tasks, with each task incorporating edge-oriented goals and obstacles. The shaded regions indicate standard errors. The $x$-axis is the CTG. The $y$-axis is the total episode return (i.e., $\sum_{k=1}^{K} G(\tau_k)$) and the maximum episode cost (i.e., $\max_{k \in K} G_c(\tau_k)$) with the corresponding CTG. The policy from safe reinforcement pretraining succeeds in converting a higher cost limit (i.e., a higher CTG) to a higher return while the policy from safe supervised pretraining fails to do so.

Supervised pretraining depends on a long replay buffer to learn from logged learning histories. Due to context length constraints, we need to subsample trajectories from these learning histories for supervised pretraining. These histories are sampled from distinct learning phases: early (low return, high cost), middle (average return, average cost), and expert (high return, low cost). To make sequence modeling feasible, trajectories need to be labeled with RTG, which provides information about the expected phase (Dai et al., 2024). However, this approach requires significantly more data, as noted by Liu et al. (2023). In contrast, reinforcement pretraining leverages online interaction histories, eliminating the need to distinguish between different data phases.

**Ablations.** We further conduct ablation studies on reinforcement pretraining and supervised pretraining, examining shared factors such as context length and model size, alongside specific factors such as the dataset size for supervised pretraining. Our findings show distinct sensitivities to these factors. Reinforcement pretraining is largely unaffected by model size, whereas model size significantly influences supervised pretraining performance. For context length, reinforcement pretraining performs better with longer sequences, while supervised pretraining performs worse. Conversely, supervised pretraining excels with shorter context lengths, while reinforcement pretraining performs poorly. These results suggest that learning long-term credit assignment is more challenging in offline reinforcement learning due to dataset constraints, whereas online learning, with access to environment interactions, handles long-term credit assignment more effectively. We confirm the well-established finding that supervised pretraining is highly sensitive to dataset size. When the dataset is small, the model fails to learn, exhibiting random behavior on out-of-distribution data. In Figure 5.(c), we compare EPPO to a naive primal–dual solver for (6) with per-episode multipliers $\lambda$. The overall structure of this naive algorithm resembles RCPO (Tessler et al., 2018). EPPO trains more stably with minimal tuning and solves SafeDarkRoom and SafeDarkMujoco without changes beyond time-limit related settings (Table 1). The detailed evaluation curves of the ablation study are provided in Figures 5 & 6 in Appendix E. We evaluate a variant of safe supervised pretraining incorporating algorithm distillation with noise in Appendix D. The results indicate that learning histories generated by noisy actions do not effectively represent true learning histories in complex environments with safety constraints.

## 5 RELATED WORKS

**ICRL.** Learning to improve on a new MDP by interacting with it and without parameter updates is first studied in meta RL (Duan et al., 2016; Wang et al., 2016). See Beck et al. (2023) for a detailed survey of meta RL. In general, the demonstrated zero parameter update generalization is weak in meta RL literature. Most of them rely on task identification, i.e., identifying pretraining tasks that are similar to the evaluation task and acting as if the evaluation task were the pretraining tasks. Laskin et al. (2023) coined the word ICRL and demonstrated generalization to evaluation tasks that are far away from the pretraining tasks, which spurs increasing attention in ICRL. Variations of algorithm distillation have been proposed for efficient in-context reinforcement learning, including the Decision Pretrained Transformer (Lee et al., 2023) and Algorithm Distillation with Noise (Zisman et al., 2023). Both approaches aim to efficiently train transformer models for in-context learning using optimal policies. See Moeini et al. (2025) for a detailed survey of ICRL. The main novelty of this work is to first study safe ICRL within the constrained MDP framework.

**Safe RL.** Safe RL methods commonly adopt CMDP formulations to enforce compliance with safety constraints during exploration and policy optimization (García & Fernández, 2015; Gu et al., 2022; Wachi et al., 2024). Constrained Policy Optimization (CPO) serves as a foundational algorithm in this area, effectively balancing performance rewards with safety requirements (Achiam et al., 2017; Wachi & Sui, 2020). Subsequent extensions, such as Constrained Update Projection and Constrained Reward Policy Optimization, have built upon CPO to enhance theoretical guarantees and practical applicability in safe RL scenarios (Xu et al., 2021; Yang et al., 2022). Shielding with function encoders and conformal prediction has been proposed to handle unseen OOD environments (Kwon et al., 2025). However, this work emphasizes runtime adaptation rather than learning algorithms through contextual information. Reward-Constrained Policy Optimization (Tessler et al., 2018) is the closest work to EPPO, but it enforces safety only in expectation rather than at the episode level. From the offline RL perspective, the Constrained Decision Transformer (Liu et al., 2023) leverages the transformer architecture for safe RL. However, existing approaches enforce safety constraints only when the evaluation task closely matches the pretraining tasks. In contrast, our safe ICRL framework verifies safety constraints even on evaluation tasks that differ substantially from the pretraining distribution.

**Safe Meta RL.** Safe meta RL has emerged to enable fast adaptation to new tasks while enforcing safety constraints. Earlier work (Luo et al., 2021) uses a three-phase approach that meta-learns a safety critic from offline data across multiple environments, adapts it to a new environment with a smaller offline dataset, and employs the adapted safety critic with a recovery policy to ensure safe learning while minimizing constraint violations. Khattar et al. (2023) provides task-averaged regret bounds for rewards and constraints via gradient-based meta-learning. More recently, Guan et al. (2024) proposes a cost-aware context encoder that uses supervised cost relabeling and contrastive learning to infer tasks based on safety constraints. To our knowledge, all previous safe meta RL works require parameter updates to achieve safe adaptation in evaluation tasks. By contrast, our work on safe ICRL enables safe adaptation to evaluation tasks without parameter updates.

## 6 CONCLUSION

This work pioneers the study of safe ICRL, examining both reinforcement pretraining and supervised pretraining. We develop a safe supervised pretraining appraoch based on the constrained decision transformer, showcasing its capability for OOD generalization while revealing its dependence on dataset size. We introduce EPPO, a safe reinforcement pretraining approach via exact penalty policy optimization to enforce strict cost constraints within each episode. Theoretically, we prove that EPPO's fixed point satisfies strict safety constraints while maximizing rewards. We conduct a thorough empirical study of the proposed approaches in challenging benchmarks that provably demand OOD generalization (Proposition 1). Such demand for OOD generalization is missing in many benchmarks used by previous works (Laskin et al., 2023; Zisman et al., 2023; Son et al., 2025) Through those challenging benchmarks, we confirm that our pretrained safe ICRL agents adapt to not only unseen but also OOD evaluation tasks while respecting the safety constraints. We envision this work guiding the development of robust, generalizable, and safe RL algorithms for real-world applications.

ETHICS STATEMENT

This research complies with the ICLR Code of Ethics (https://iclr.cc/public/CodeOfEthics). Our study on safe in-context reinforcement learning (ICRL), conducted within the framework of constrained Markov Decision Processes, was evaluated exclusively in simulated Safe-Gym environments. No human subjects, sensitive data, or real-world deployments were involved, thereby eliminating potential ethical risks associated with such factors. Our methodology prioritizes safety by designing an ICRL agent that adheres to user-specified cost thresholds during adaptation to out-of-distribution tasks, ensuring safe behavior when required. No conflicts of interest or external funding influenced this work. We are committed to upholding the highest standards of research integrity, transparency, and reproducibility as outlined in the ICLR guidelines. All implementation details will be made publicly available in the supplementary materials.

REPRODUCIBILITY STATEMENT

To ensure reproducibility, the complete source code will be provided in https://osf.io/rs759/files/osfstorage?view_only=9810212f4bc5448b8f1caaafd334353d. Training details are documented in Appendix C, and all theoretical results, including assumptions and derivations, are presented with clear explanations in Appendix B.

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

# A  ALGORITHM

---

**Algorithm 1:** EPPO Implemented with DDPG

---

1: **Input:** discount factor $\gamma$, cost threshold $\delta$, number of training steps $T_{\max}$, batch size $N$, env time limit $t_{\max}$, episodes-per-history range $[K_{\min}, K_{\max}]$, CTG range $[\mathrm{CTG}_{\min}, \mathrm{CTG}_{\max}]$, environment distribution $\mathcal{E}$

2: **Initialize:** actor $\pi(\cdot \mid s, H, \mathrm{CTG}; \theta_p)$, reward critic $Q(s, a; \theta_v)$, cost critic $Q^c(s, a; \theta_c)$, and target nets $\{\pi', Q', Q^{c'}\}$. $\lambda \leftarrow 0$, $\quad$ replay buffer $R \leftarrow \varnothing$

3: **for** $T \leftarrow 1$ **to** $T_{\max}$ **do**

4: $\quad K \leftarrow \mathrm{rnd}(K_{\min}, K_{\max}), \quad \mathrm{CTG} \leftarrow \mathrm{rnd}(\mathrm{CTG}_{\min}, \mathrm{CTG}_{\max}), \quad t \leftarrow 0$
$\quad H \leftarrow [\,]\,/\!/$ `list of episodes`

5: $\quad$ sample $\mathrm{env} \sim \mathcal{E}, \quad s_t \leftarrow \mathrm{env.reset()}$

6: $\quad$ **for** $k \leftarrow 1$ **to** $K$ **do**

7: $\quad\quad t_{\mathrm{start}} \leftarrow t, \quad \mathrm{CTG}_k \leftarrow \mathrm{CTG}, \quad e_k \leftarrow [\,]\,/\!/$ `list of transitions`

8: $\quad\quad$ **while** $t - t_{\mathrm{start}} < t_{\max}$ **and** $s_t$ not terminal **do**

9: $\quad\quad\quad$ sample $a_t \sim \pi_{\theta_p}(\cdot \mid s_t, H, \mathrm{CTG}_k)$

10: $\quad\quad\quad$ step $a_t$ in env $\rightarrow (s_{t+1}, r_{t+1}, c_{t+1})$

11: $\quad\quad\quad$ append $(s_t, a_t, r_{t+1}, c_{t+1}, s_{t+1}, \mathrm{CTG}_k)$ to $e_k$

12: $\quad\quad\quad \mathrm{CTG}_k \leftarrow \mathrm{CTG}_k - c_{t+1}, \quad t \leftarrow t + 1$

13: $\quad\quad$ **end while**

14: $\quad\quad$ append $e_k$ to $H$

15: $\quad\quad s_t \leftarrow \mathrm{env.reset()}$

16: $\quad$ **end for**

17: $\quad$ append $H$ to $R$

18: $\quad \mathcal{D} \leftarrow \mathrm{Sample}(R, N)\,/\!/\ N$ `trajectories`

19: $\quad$ reset accumulators: $d\theta_v \leftarrow 0, d\theta_c \leftarrow 0, d\theta_p \leftarrow 0, d\lambda \leftarrow 0$

20: $\quad$ **for each** trajectory $H \in \mathcal{D}$ **do**

21: $\quad\quad$ **for each** episode $e \in H$ **do**

22: $\quad\quad\quad C_e \leftarrow \sum_{c \in e} c, \quad C_e^{\max} \leftarrow -\infty, \quad v_e \leftarrow \mathbf{1}\{C_e > \delta\}$

23: $\quad\quad\quad$ **for each** $(s_t, a_t, r_{t+1}, c_{t+1}, s_{t+1}, \mathrm{CTG}_t) \in e$ **do**

24: $\quad\quad\quad\quad$ sample $a'_{t+1} \sim \pi_{\theta'_p}(\cdot \mid s_{t+1}, H_{\leq t+1}, \mathrm{CTG}_t - c_{t+1})$

25: $\quad\quad\quad\quad L_v \leftarrow \big(Q_{\theta_v}(s_t, a_t) - [\, r_{t+1} + \gamma Q_{\theta'_v}(s_{t+1}, a'_{t+1})\,]\big)^2$

26: $\quad\quad\quad\quad d\theta_v \leftarrow d\theta_v + \nabla_{\theta_v} L_v$

27: $\quad\quad\quad\quad L_c \leftarrow \big(Q^c_{\theta_c}(s_t, a_t) - [\, c_{t+1} + \gamma Q^c_{\theta'_c}(s_{t+1}, a'_{t+1})\,]\big)^2$

28: $\quad\quad\quad\quad d\theta_c \leftarrow d\theta_c + \nabla_{\theta_c} L_c$

29: $\quad\quad\quad\quad$ sample $\hat{a}_t \sim \pi_{\theta_p}(\cdot \mid s_t, H_{\leq t}, \mathrm{CTG}_t)$

30: $\quad\quad\quad\quad L_p \leftarrow -Q_{\theta_v}(s_t, \hat{a}_t) + \lambda\, v_e\, Q^c_{\theta_c}(s_t, \hat{a}_t)$

31: $\quad\quad\quad\quad d\theta_p \leftarrow d\theta_p + \nabla_{\theta_p} L_p$

32: $\quad\quad\quad$ **end for**

33: $\quad\quad\quad C_e^{\max} \leftarrow \max\{C_e^{\max}, C_e\}$

34: $\quad\quad$ **end for**

35: $\quad\quad d\lambda \leftarrow d\lambda + C_e^{\max}$

36: $\quad$ **end for**

37: $\quad$ apply parameter updates to $\theta_v, \theta_c, \theta_p, \lambda$ using $d\theta_v, d\theta_c, d\theta_p, d\lambda$

38: $\quad$ update targets $\pi', Q', Q^{c'}$

39: **end for**

---

Note that our proposed optimization (7) can be used with any policy optimizer. In our implementation, we chose DDPG following Grigsby et al. (2024a) to support continuous and discrete action spaces and to enable highly parallel data-collection while doing off-policy training.

# B  PROOFS

We organize the proofs into two subsections: one for Theorem 1 and one for Proposition 1. For Theorem 1, we first prove the supporting lemma and then Theorem 1 itself. For Proposition 1, we provide proofs for both Total Variation Distance and KL Divergence.

## B.1 PROOF OF THEOREM 1

**Lemma 1.** *Let* $[x]_+ = \max\{0, x\}$. *Fix* $(\bar\lambda, \bar{s}) \in \mathbb{R} \times \mathbb{R}$. *If for all sufficiently small* $\eta > 0$, $\bar\lambda = [\bar\lambda + \eta \bar{s}]_+$, *then* $\bar\lambda \geq 0$, $\bar{s} \leq 0$, *and* $\bar\lambda \bar{s} = 0$.

*Proof.* Consider two cases.

*Case* $\bar\lambda > 0$. For all small $\eta > 0$, $\bar\lambda + \eta\bar{s} > 0$, hence

$$[\bar\lambda + \eta\bar{s}]_+ = \bar\lambda + \eta\bar{s}.$$

The equality $\bar\lambda = [\bar\lambda + \eta\bar{s}]_+$ then forces $\bar\lambda = \bar\lambda + \eta\bar{s}$, so $\bar{s} = 0$. Thus $\bar{s} \leq 0$ and $\bar\lambda\bar{s} = 0$ hold.

*Case* $\bar\lambda = 0$. Then we have $0 = [\eta\bar{s}]_+ = \max\{0, \eta\bar{s}\}$ for all small $\eta > 0$, which implies $\eta\bar{s} \leq 0$, hence $\bar{s} \leq 0$. Trivially $\bar\lambda\bar{s} = 0$ and $\bar\lambda \geq 0$. $\qquad\square$

We now proceed to the proof of Theorem 1.

*Proof.* Let $J(\pi) := \mathbb{E}_\pi\left[\sum_{k=1}^K G(\tau_k)\right]$ in this proof. We prove the theorem in both directions. Also let $g_k^+(\pi) := \max\{0, g_k(\pi)\}$, and $s(\pi) := \max_k g_k(\pi)$.

*1. Fixed point $\Rightarrow$ Primal optimal.*

*Feasibility.* By the lemma 1, $s(\bar\pi) \leq 0$. Since $s(\bar\pi) = \max_i g_i(\bar\pi)$, each $g_i(\bar\pi) \leq 0$. Thus $\bar\pi$ is feasible.

*Optimality.* Let $\pi^\star$ be any optimal feasible policy (exists by assumption 1). On the feasible set, $g_i^+(\pi^*) = 0$, so $L_\Sigma(\pi^*, \bar\lambda) = J(\pi^*)$. Because $\bar\pi$ maximizes $L_\Sigma(\cdot, \bar\lambda)$,

$$J(\bar\pi) = L_\Sigma(\bar\pi, \bar\lambda) \geq L_\Sigma(\pi^\star, \bar\lambda) = J(\pi^\star).$$

Conversely, by optimality of $\pi^\star$ among feasible policies and feasibility of $\bar\pi$, $J(\bar\pi) \leq J(\pi^\star)$. Hence $J(\bar\pi) = J(\pi^\star)$.

*2. Primal optimal $\Rightarrow$ Fixed point.*

Let $\lambda^\star$ be an optimal dual solution (assumption 2). Then by strong duality,

$$p^\star = \max_\pi \left(J(\pi) - \sum_i \lambda_i^\star g_i(\pi)\right).$$

Therefore, for every $\pi$,

$$J(\pi) - \sum_i \lambda_i^\star g_i(\pi) \leq p^\star. \tag{8}$$

Using $\lambda_i^\star \leq \|\lambda^\star\|_\infty$ and $g_i^+(\pi) \geq g_i(\pi)$,

$$\sum_i \lambda_i^\star g_i(\pi) \leq \|\lambda^\star\|_\infty \sum_i g_i^+(\pi).$$

Subtract this from $J(\pi)$ and combine with (8):

$$L_\Sigma\left(\pi, \|\lambda^\star\|_\infty\right) = J(\pi) - \|\lambda^\star\|_\infty \sum_i g_i^+(\pi) \leq J(\pi) - \sum_i \lambda_i^\star g_i(\pi) \leq p^\star. \tag{9}$$

For a primal-optimal $\pi^\star$, feasibility gives $\sum_i g_i^+(\pi^\star) = 0$, hence

$$L_\Sigma\big(\pi^\star, \|\lambda^\star\|_\infty\big) = J(\pi^\star) = p^\star.$$

Together with (9), this shows

$$\pi^\star \in \arg\max_\pi L_\Sigma\big(\pi, \|\lambda^\star\|_\infty\big).$$

It remains to check multiplier stationarity at $\bar\lambda = \|\lambda^\star\|_\infty$. By complementary slackness, $\sum_i \lambda_i^\star g_i(\pi^\star) = 0$. If $\|\lambda^\star\|_\infty > 0$, at least one constraint is active at $\pi^\star$, so $\max_i g_i(\pi^\star) = 0$. Hence $[\bar\lambda + \eta\, s(\pi^\star)]_+ = [\bar\lambda + \eta \cdot 0]_+ = \bar\lambda$ for all small $\eta > 0$. If $\|\lambda^\star\|_\infty = 0$, $\pi^\star$ is unconstrained-optimal with $s(\pi^\star) \le 0$, and $[0 + \eta\, s(\pi^\star)]_+ = 0$. In both cases the projection is stationary. Thus $(\pi^\star, \bar\lambda)$ with $\bar\lambda = \|\lambda^\star\|_\infty$ is a fixed point.

*On the multiplier update.*
If $s(\pi_{t+1}) > 0$, then $\lambda_{t+1} = [\lambda_t + \eta s(\pi_{t+1})]_+ \ge \lambda_t$. If constraints are strict ($s(\pi_{t+1}) < 0$), then $\lambda_{t+1} \le \lambda_t$. Under persistent violation, $\lambda_t$ eventually exceeds $\|\lambda^*\|_\infty$, by the exact-penalty bound (9), any maximizer of $L_\Sigma(\cdot, \lambda_t)$ is then feasible/optimal and $s(\pi_{t+1}) \le 0$, so $\lambda$ stabilizes. □

Note: The full convergence proof to an exact fixed point is beyond the scope of this work.

### B.2 PROOF OF PROPOSITION 1

**Total Variation Distance** First, we present the exact form of probabillites for $\mathbb{P}_{\text{train}}(O)$ and $\mathbb{P}_{\text{test}}(O)$ where $O \in \mathcal{O} = \{(i,j)\}_{1 \le i \le n, 1 \le j \le m}$. Let the map of center $c = (i_c, j_c)$. Then, $\mathbb{P}_{\text{train}}((i,j)) \propto e^{-\alpha d((i,j),c)}$ and $\mathbb{P}_{\text{train}}((i,j)) \propto e^{-\alpha d((i,j),c)}$ implies:

$$\mathbb{P}_{\text{train}}((i,j)) = \frac{e^{-\alpha((i-i_c)^2+(j-j_c)^2)}}{Z_{-\alpha}} \quad \text{and} \quad \mathbb{P}_{\text{test}}((i,j)) = \frac{e^{\alpha((i-i_c)^2+(j-j_c)^2)}}{Z_\alpha},$$

where $Z_{-\alpha} = \sum_{(i',j')\in\mathcal{O}} e^{-\alpha((i'-i_c)^2+(j'-j_c)^2)}$ and $Z_\alpha = \sum_{(i',j')\in\mathcal{O}} e^{\alpha((i'-i_c)^2+(j'-j_c)^2)}$. Let us define $supp(P)$ as the set $\{x \in Dom(P) \mid P(x) > 0\}$. If $supp(\mathbb{P}_{\text{train}}) \cap supp(\mathbb{P}_{\text{test}}) = \emptyset$ holds, then for each $O \in \mathcal{O}$, $\mathbb{P}_{\text{test}}(O) > 0$ implies $\mathbb{P}_{\text{train}}(O) = 0$, and $\mathbb{P}_{\text{train}}(O) > 0$ implies $\mathbb{P}_{\text{test}}(O) = 0$. Hence, we obtain

$$\delta(\mathbb{P}_{\text{train}}, \mathbb{P}_{\text{test}}) = \frac{1}{2} \sum_{O\in\mathcal{O}} |\mathbb{P}_{\text{train}}(O) - \mathbb{P}_{\text{test}}(O)| \tag{10}$$

$$= \frac{1}{2} \sum_{O\in\mathcal{O}} (|\mathbb{P}_{\text{train}}(O)| + |\mathbb{P}_{\text{test}}(O)|) = 1. \tag{11}$$

Now, we claim that $supp(\mathbb{P}_{\text{train}}) \cap supp(\mathbb{P}_{\text{test}}) = \emptyset$ as $\alpha \to \infty$. For the training distribution $\mathbb{P}_{\text{train}}$, the term $e^{-\alpha\big((i-i_c)^2+(j-j_c)^2\big)} \to 0$ if $(i - i_c)^2 + (j - j_c)^2 > 0$. Hence,

$$Z_{-\alpha} = e^{-\alpha\cdot 0} + \sum_{(i,j)\neq(i_c,j_c)} e^{-\alpha\big((i-i_c)^2+(j-j_c)^2\big)} \to 1$$

Hence, only when $i = i_c$ and $j = j_c$, we have positive probability $\mathbb{P}_{\text{train}}((i_c, j_c)) = \frac{e^{-\alpha\cdot 0}}{Z_{-\alpha}} = \frac{1}{Z_{-\alpha}} = 1$. Thus $supp(\mathbb{P}_{\text{train}}) \to \{(i_c, j_c)\}$

For the test distribution $\mathbb{P}_{\text{test}}$, the term $e^{\alpha\big((i-i_c)^2+(j-j_c)^2\big)}$ is maximized when $(i - i_c)^2 + (j - j_c)^2$ is maximized. Let $d_{\max} = \max_{(i,j)\in\mathcal{O}} \big((i - i_c)^2 + (j - j_c)^2\big)$, and $\mathcal{O}_{\text{edge}} = \big\{(i,j) \in \mathcal{O} \mid (i - i_c)^2 + (j - j_c)^2 = d_{\max}\big\}$. Then the partial function can be decomposed into two terms:

$$Z_\alpha = \sum_{(i,j)\in\mathcal{O}_{\text{edge}}} e^{\alpha d_{\max}} + \sum_{(i,j)\notin\mathcal{O}_{\text{edge}}} e^{\alpha\big((i-i_c)^2+(j-j_c)^2\big)}.$$

Hence, as $\alpha \to \infty$, $(i - i_c)^2 + (j - j_c)^2 < d_{\max}$ implies that $\frac{e^{\alpha((i-i_c)^2+(j-j_c))}}{Z_\alpha} \to 0$. Thus, we obtain

$$\mathbb{P}_{\text{test}}\left((i,j)\right) \to \begin{cases} \frac{1}{|\mathcal{O}_{\text{edge}}|} & \text{if } (i,j) \in \mathcal{O}_{\text{edge}}, \\ 0 & \text{otherwise} \end{cases} \quad as \quad \alpha \to \infty.$$

Hence, $supp(\mathbb{P}_{\text{test}}) \to \mathcal{O}_{\text{edge}}$. Finally, we conclude that $supp(\mathbb{P}_{\text{train}}) \cap supp(\mathbb{P}_{\text{test}}) = \{(i_c, j_c)\} \cap \mathcal{O}_{\text{edge}} = \emptyset$ as $\alpha \to \infty$.

**KL Divergence**    We keep the same notation to the proof so far. By the definition of KL divergence and explicit form of the probabilities, we have

$$D_{\text{KL}}\left(\mathbb{P}_{\text{train}} \| \mathbb{P}_{\text{test}}\right) = \sum_{(i,j) \in \mathcal{O}} \mathbb{P}_{\text{train}}\left((i,j)\right) \log \frac{\mathbb{P}_{\text{train}}\left((i,j)\right)}{\mathbb{P}_{\text{test}}\left((i,j)\right)} \tag{12}$$

$$= \sum_{(i,j) \in \mathcal{O}} \mathbb{P}_{\text{train}}\left((i,j)\right) \log \frac{Z_\alpha e^{-\alpha((i-i_c)^2+(j-j_c)^2)}}{Z_{-\alpha} e^{\alpha((i-i_c)^2+(j-j_c)^2)}} \tag{13}$$

$$= \sum_{(i,j) \in \mathcal{O}} \mathbb{P}_{\text{train}}\left((i,j)\right) \left( \log \frac{Z_\alpha}{Z_{-\alpha}} - 2\alpha((i-i_c)^2 + (j-j_c)^2) \right). \tag{14}$$

As we have shown in the previous proof for TV distance, $Z_\alpha \to |\mathcal{O}_{\text{edge}}|e^{\alpha d_{\max}}$ and $Z_{-\alpha} \to 1$. Hence, for pairs $(i,j) \neq (i_c, j_c)$, the associated term $\mathbb{P}_{\text{train}}\left((i,j)\right) \left( \log \frac{Z_\alpha}{Z_{-\alpha}} - 2\alpha((i-i_c)^2 + (j-j_c)^2) \right)$ goes to $\mathbb{P}_{\text{train}}\left((i,j)\right) \left( \log |\mathcal{O}_{\text{edge}}| + \alpha d_{\max} - 2\alpha d((i,j),c) \right)$. For $(i,j) \neq (i_c, j_c)$ and $d_{\min} = \min_{(i,j) \neq (i_c,j_c)} d((i,j),c)$, we have

$$P_{\text{train}}\left((i,j)\right) = \frac{e^{-\alpha d((i,j),c)}}{Z_{-\alpha}} \leq e^{-\alpha d_{\min}} \to 0. \tag{15}$$

Since the exponential rate converges to zero more rapidly than the linear rate with respect to $\alpha$, the term $\mathbb{P}_{\text{train}}\left((i,j)\right) \left( \log |\mathcal{O}_{\text{edge}}| + \alpha d_{\max} - 2\alpha d((i,j),c) \right)$ goes to 0 as $\alpha \to \infty$. Therefore, we only consider the term when $(i,j) = (i_c, j_c)$. But this term comes down to $\mathbb{P}_{\text{train}}((i,j)) \log \frac{Z_\alpha}{Z_{-\alpha}}$, which goes to $\infty$ as $\alpha \to \infty$. Thus $D_{\text{KL}}\left(\mathbb{P}_{\text{train}} \| \mathbb{P}_{\text{test}}\right) \geq \infty$.

## C  TRAINING DETAILS

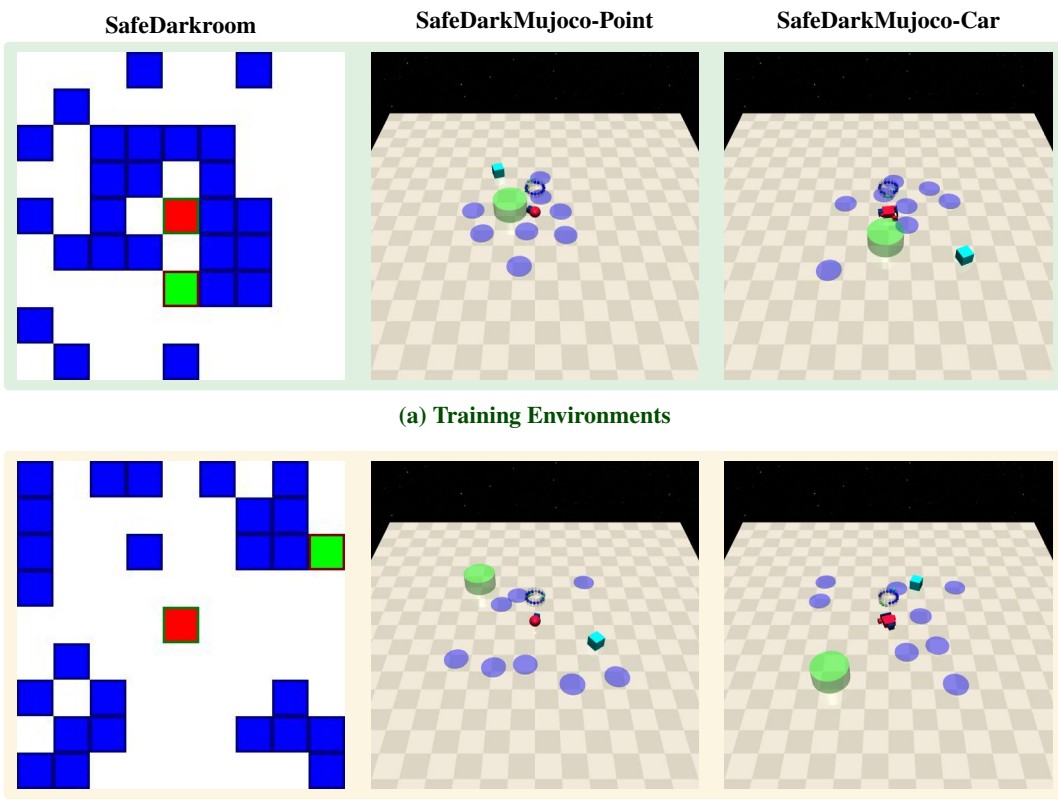

**(a) Training Environments**

**(b) Test Environments**

Figure 3: During training, goals and obstacles are generated with a center-oriented approach, while during evaluation, they are edge-oriented. This applies consistently to both goals and obstacles. We set $\alpha = 0.5$ for generating goals and obstacles. The red color denotes the robot, the green color represents the goal location, and obstacles are depicted in shades of blue.

**Environments.**  We introduce the details of the environments. The environments are visualized in Figure 3. Both environment uses $\alpha = 0.5$ to generate goals and obstacles.

For SafeDarkRoom, We use a 9x9 Grid and give one reward upon reaching the goal and incur one cost when going on an obstacle cell. We terminate the episode whenever the agent reaches the goal, which results in a truly sparse reward, often referred to as *DarkRoom Hard*.

For SafeDarkMujoco, a continuous state and action space counterpart to SafeDarkRoom, the agent lacks lidar information and is blind to goal and obstacle locations. Instead, it perceives its own position and rotation matrix. A sparse reward is obtained when the agent reaches the goal, terminating the episode. To enhance runtime efficiency, we employ macro actions, compressing $n$ simulation steps into a single step. For example, with $n = 5$, a policy action is repeated over five internal simulation steps, reducing the default 250 simulation steps to $\frac{250}{n}$. In our experiments, we set $n = 5$.

**Reinforcement Pretraining.**  While running Algorithm 1 for reinforcement pretraining, we resample a new environment from the training distribution described in Section 4 every $K$ episodes. For each environment, a CTG is also sampled, ranging from $[1, 15]$ for SafeDarkRoom and $[10, 50]$ for SafeDarkMujoco. The remaining hyperparameters are provided in Table 1.

Our architecture follows Grigsby et al. (2024b). We employ an MLP time-step encoder that maps each tuple $(S_t, A_t, R_t, C_t)$ to an embedding, which is then fed into a transformer-based trajectory encoder. A prediction head outputs either the action distribution (for discrete actions) or the value (for continuous actions).

In practice, since the environments are generated randomly, it is possible that an environment doesn't have a feasible solution, causing the agent to inevitably violate the cost threshold. This results in $\lambda$ growing too large in an attempt to mitigate these cases. To avoid this, we suggest capping $\lambda$ to a reasonably large value. Ignoring abnormally high-cost episodes is another solution worth exploring.

| Parameter | SafeDarkRoom | SafeDarkMujoco-Point | SafeDarkMujoco-Car |
|---|---|---|---|
| $K_{\min}, K_{\max}$ | 50, 50 | 20, 20 | 20, 20 |
| Episode time limit $t_{\max}$ | 30 | 75 | 75 |
| Replay buffer capacity | 100,000 | 100,000 | 100,000 |
| Embedding Dim | 64 | 64 | 64 |
| Hidden Dim | 64 | 64 | 64 |
| Num Layers | 4 | 4 | 4 |
| Num Heads | 8 | 8 | 8 |
| Seq Len | 1500 | 1500 | 1500 |
| Attention Dropout | 0 | 0 | 0 |
| Residual Dropout | 0 | 0 | 0 |
| Embedding Dropout | 5 | 5 | 5 |
| Learning Rate | 3e-4 | 3e-4 | 3e-4 |
| Betas | (0.9, 0.99) | (0.9, 0.99) | (0.9, 0.99) |
| Clip Grad | 1.0 | 1.0 | 1.0 |
| Batch Size | 32 | 32 | 32 |
| Num Updates | 30k | 10k | 10k |
| Optimizer | Adam | Adam | Adam |

Table 1: Parameters for Safe Reinforcement Pretraining

**Supervised Pretraining.** In supervised pretraining, we collect a dataset $\mathcal{D} = \{\Xi_i\}$ comprising multiple trajectories, where each trajectory $\Xi_i \doteq (\tau_1, \tau_2, \ldots, \tau_K)$ represents a sequence of episodes generated by running existing safe RL algorithms on various CMDPs. Each episode $\tau_k$ in a trajectory $\Xi_i$ comprises states, actions, rewards, and costs, with the episode return $G(\tau_k) \doteq \sum_{t=1}^{T} R_t$ and cost-to-go $G_{c,t}(\tau_k) \doteq \sum_{i=t+1}^{T} C_i$, expected to increase and decrease, respectively, with $k$ as the RL algorithm learns. The policy $\pi_\theta$ is trained autoregressively using an imitation learning loss to distill the behavior of RL algorithms demonstrated in the dataset, following the concept of algorithm distillation (Laskin et al., 2023). For discrete action spaces (e.g., SafeDarkRoom), we use a cross-entropy loss, defined as:

$$\mathcal{L}_{\mathrm{CE}}(\theta) = \mathbb{E}_{\Xi_i \sim \mathcal{D}} \left[ -\log \pi_\theta(A_t^k \mid S_t^k, H_t^k, G_t(\tau_k), G_{c,t}(\tau_k)) \right], \tag{16}$$

where $\pi_\theta$ is the categorical distribution over discrete actions. For continuous action spaces (e.g., SafeDarkMujoco), we use an L2 loss, where the transformer outputs an action mean $\mu_t = \mu_\theta(S_t^k, H_t^k, G_t(\tau_k), G_{c,t}(\tau_k))$, minimizing:

$$\mathcal{L}_{\mathrm{L2}}(\theta) = \mathbb{E}_{\Xi_i \sim \mathcal{D}} \left[ \left\| A_t^k - \mu_\theta(S_t^k, H_t^k, G_t(\tau_k), G_{c,t}(\tau_k)) \right\|^2 \right]. \tag{17}$$

These losses extend the imitation learning loss from Equation (5) by incorporating RTG and CTG conditioning, aligning with our goal of distilling RL algorithms into the policy's forward pass while optimizing for safety constraints.

**Dataset Collection.** For supervised pretraining, we collect learning trajectories using reinforcement learning (RL) algorithms. As our base RL algorithm, we employ PPO-Lagrangian Schulman et al. (2017); Ray et al. (2019), which is designed to maximize rewards while enforcing safety constraints. These trajectories capture behaviors that learn to avoid obstacles.

To introduce variation in the learned behaviors, we vary the cost limits in PPO-Lag across multiple settings. For the SafeDarkRoom environment, we use three cost limits: 0, 2.5, and 5.0. For the SafeDarkMujoco environment, we similarly use three cost limits: 0, 5, and 10. For each cost limit, we collect 50,000 steps of learning history in both environments.

**Hyper Parameters.** We report the hyperparameters used for reinforcement pretraining (Table 1) and supervised pretraining (Table 2). Reinforcement pretraining adopts the AMAGO framework (Grigsby

et al., 2024a) integrated with our novel EPPO method. Supervised pretraining employs a constrained decision transformer as the backbone (Liu et al., 2023).

| Parameter | SafeDarkRoom | SafeDarkMujoco-Point | SafeDarkMujoco-Car |
|---|---|---|---|
| Embedding Dim | 64 | 64 | 64 |
| Hidden Dim | 512 | 256 | 256 |
| Num Layers | 8 | 8 | 8 |
| Num Heads | 8 | 8 | 8 |
| Seq Len | 100 | 300 | 200 |
| Attention Dropout | 0.5 | 0.5 | 0.5 |
| Residual Dropout | 0.1 | 0.1 | 0.1 |
| Embedding Dropout | 0.3 | 0.3 | 0.3 |
| Learning Rate | 3e-4 | 3e-4 | 3e-4 |
| Betas | (0.9, 0.99) | (0.9, 0.99) | (0.9, 0.99) |
| Clip Grad | 1.0 | 1.0 | 1.0 |
| Batch Size | 512 | 128 | 128 |
| Num Updates | 300k | 500k | 500k |
| Optimizer | Adam | Adam | Adam |

Table 2: Parameters for Safe Supervised Pretraining

# D  VARIATION OF SUPERVISED PRETRAINING

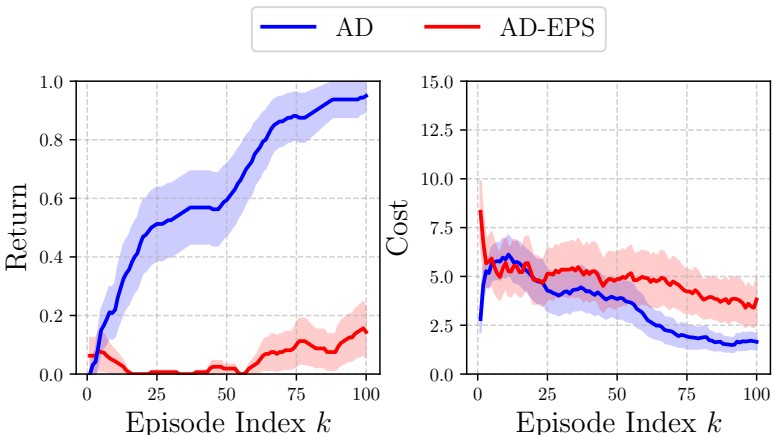

Figure 4: Performance Comparison of Algorithm Distillation (AD) style supervised pretraining and Algorithm Distillation with Noise (AD-EPS) style supervised pretraining in SafeDarkRoom. AD-EPS fails to generalize in OOD environments.

In this section, we compare the performance of Algorithm Distillation (AD) (Laskin et al., 2023) and Algorithm Distillation with Noise (AD-EPS) (Zisman et al., 2023) in SafeDarkRoom. AD-EPS is designed to learn in-context RL algorithms using datasets generated from a perturbed optimal policy. This approach allows efficient generation of learning trajectories from a single optimal policy. However, our results in complex environments with cost signals reveal that AD-EPS relies on artificial trajectories. AD-EPS fails to learn effective in-context RL algorithms when relying solely on perturbed optimal policies, as the perturbations do not account for obstacle avoidance in SafeDarkRoom environment and instead introduce random action variations.

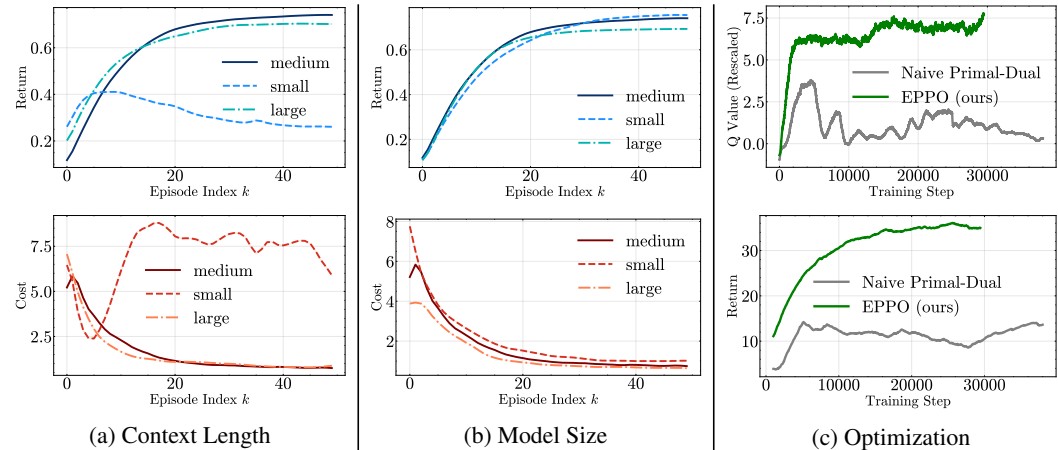

(a) Context Length      (b) Model Size      (c) Optimization

Figure 5: **Ablations of Safe Reinforcement Pretraining on SafeDarkRoom.** **(a), (b)** The evaluation is set up similarly to Question (i). For context length, we compare 150 and 3000 against the base value of 1500. For model size, we compare embedding dimensions 32 and 128 against the base of 64. **(c)** At each training step, the average total return of 50 episodes across 10 random test environments, and the average Q Value across 50 episodes of 250 random train enironments are plotted. EPPO is easier to tune and more stable to train than the naive primal-optimal method.

# E   ABLATION STUDIES

In this section, we present our ablation studies on reinforcement pretraining and supervised pretraining, examining shared factors such as context length and model size, as well as specific factors like dataset size for supervised pretraining. Our findings reveal distinct sensitivities to these factors. Reinforcement pretraining remains largely unaffected by model size, whereas supervised pretraining performance is significantly influenced by model size (See Figures 5.(b) and 6.(b)). Regarding context length, reinforcement pretraining benefits from longer sequences, while supervised pretraining exhibits degraded performance with longer contexts. Conversely, supervised pretraining performs better with shorter context lengths, where reinforcement pretraining struggles (See Figures 5.(a) and 6.(a)). These results indicate that learning long-term credit assignment is more challenging in offline reinforcement learning due to dataset constraints, whereas online learning, with access to environment interactions, manages long-term credit assignment more effectively. We also confirm the well-established finding that supervised pretraining is highly sensitive to dataset size, with models failing to learn and exhibiting random behavior on out-of-distribution data when the dataset is small (See Figure 6.(c)).

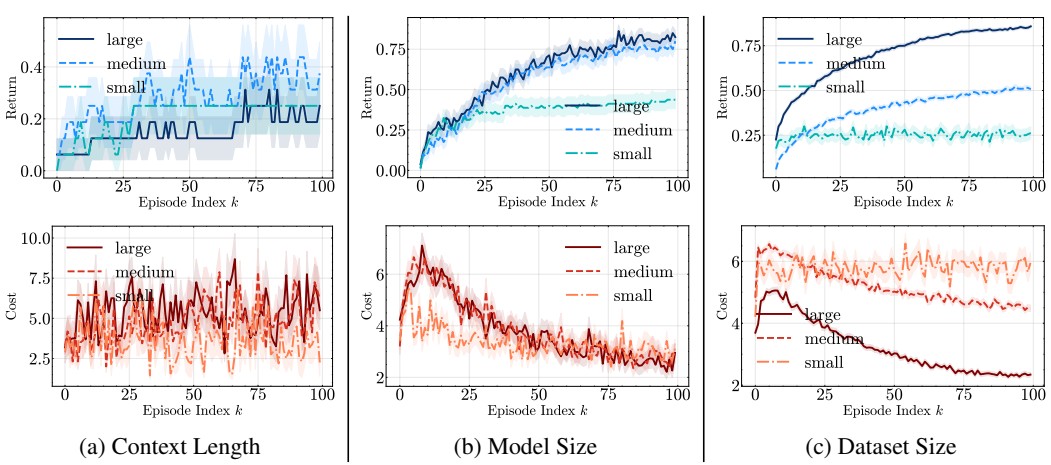

(a) Context Length          (b) Model Size          (c) Dataset Size

Figure 6: **Ablations of Safe Supervised Pretraining on SafeDarkRoom.** The evaluation is set up similarly to Question (i). For context length, we use a smaller base model and test three sequence lengths: 100, 500, and 1000. For dataset size, large refers to the full dataset, medium uses 50% of the dataset, and small uses 5% of the original dataset. Model size increases with the number of hidden layers: 2, 4, and 8, keeping other factors constant.

