# OpenReview forum: "Safe In-Context Reinforcement Learning"
_ICLR.cc/2026/Conference — Submitted to ICLR 2026_

### Official Review · Reviewer_aqYG · 2025-10-15

**Soundness:** 2
**Presentation:** 3
**Contribution:** 2
**Rating:** 2
**Confidence:** 4

**Summary:**

This work presents the first approach to achieving safe ICRL under CMDP, enabling safe adaptation to OOD scenarios without requiring parameter updates. The proposed method first performs safe supervised pretraining in an offline manner, followed by further refinement through EPPO in the online environment. Experiments conducted in the SafeDarkRoom and SafeDarkMujoco environments demonstrate the method’s strong OOD adaptation capability.

**Strengths:**

- This work investigates a novel problem setting, Safe ICRL, and proposes a principled solution to address it.
- To account for the unique characteristics of ICRL, where multiple trajectories are used as history, the authors improve upon the conventional primal–dual framework by proposing the EPPO algorithm, which updates the Lagrange multiplier based on the maximum cost observed across multiple trajectories.
- Experiments conducted in manually designed environments demonstrate the method’s ability to achieve robust OOD adaptation.

**Weaknesses:**

- The authors should more clearly and explicitly motivate the study of this problem. From my perspective, Safe ICRL does not appear to be a compelling research problem. First, the authors use a “dark” setup in their experimental environments—turning off all radar and local observation information and only retaining the agent’s own state—and create OOD scenarios by changing obstacle and target locations under this restricted setting. However, I doubt such tasks occur in real-world applications, since sensors like radar are typically available. With radar, changes in obstacle distribution might not constitute true OOD scenarios. Second, ICRL requires continual exploration and trajectory collection to adapt to new environments. While this is acceptable in standard RL, in Safe RL such exploration repeatedly generates new unsafe interactions, which can be risky.
- The motivation for the proposed method is insufficiently clear. The stated goal of Safe ICRL is to enable more efficient adaptation to OOD scenarios, yet the main technical contribution appears to be online reinforcement pretraining—a technique that is common in online safe RL. The authors should more thoroughly explain why reinforcement pretraining improves OOD adaptation. Is it because the online phase yields more or higher-quality safe trajectories for training? If the benefit is merely due to increased quantity or quality of data, then the claimed motivation rooted in ICRL may be inconsistent.
- The experimental environments are relatively simple and too few. The authors should evaluate on more tasks in SafetyGymnasium, e.g., construct multi-task scenarios in MuJoCo Velocity tasks by varying safe velocity thresholds, etc.
- The experimental baselines are insufficient.
    - The supervised pretraining baseline is purely offline, whereas the proposed method is offline + online; the settings and amounts of training data therefore differ, making the comparison unfair and insufficient to demonstrate the advantage of the proposed method.
    - The paper does not adequately compare EPPO with other online safe RL algorithms (e.g., RCPO [1], CVPO [2], CAL [3], RESPO [4]); only a comparison with a naive primal–dual method in the appendix for a single environment is shown, and cost results are not provided, which is unconvincing.
    - The authors should also compare with more recent baselines in the ICRL literature to demonstrate the effectiveness of their cost-oriented design in the safe setting.
    - Additionally, in OOD experiments, comparisons should be made to meta-safe-RL methods under consistent offline/online settings.

[1] [Reward constrained policy optimization](https://arxiv.org/abs/1805.11074)

[2] [Constrained variational policy optimization for safe reinforcement learning](https://proceedings.mlr.press/v162/liu22b.html)

[3] [Off-policy primal-dual safe reinforcement learning](https://arxiv.org/abs/2401.14758)

[4] [Iterative reachability estimation for safe reinforcement learning](https://proceedings.neurips.cc/paper_files/paper/2023/hash/dca63f2650fe9e88956c1b68440b8ee9-Abstract-Conference.html)

**Questions:**

Were the reported cost values in the experiments normalized? If yes, what kind of normalization was used? If not, given that the cost values are already small in the zero-shot setting, does this indicate that the experimental environments are too simple in terms of safety constraints?

---

> ### Author Response · Authors · 2025-11-22
>
> Thank you for your detailed review. We have provided the following response:
>
> > First, the authors use a “dark” setup in their experimental environments—turning off all radar and local observation information and only retaining the agent’s own state—and create OOD scenarios by changing obstacle and target locations under this restricted setting. However, I doubt such tasks occur in real-world applications, since sensors like radar are typically available. With radar, changes in obstacle distribution might not constitute true OOD scenarios.
>
> This setup is standard in the ICRL literature [1-4], and for good reason: our goal is to test an agent’s ability to adapt to a new MDP through interaction, not to rely on perfect state observability. Partial observability is the common and often realistic regime for evaluating exploration and adaptation, and the “dark” setting is simply an extreme but clean POMDP that stresses these capabilities.
>
> > Second, ICRL requires continual exploration and trajectory collection to adapt to new environments. While this is acceptable in standard RL, in Safe RL such exploration repeatedly generates new unsafe interactions, which can be risky.
>
> This is exactly the safety gap our paper targets: we design safe ICRL so that adaptation via interaction does not require repeated unsafe exploration. Empirically, during OOD adaptation the reinforcement-pretrained agent maintains episode costs within the user-specified limit/CTG (see max-cost curves in Figure 2), while still improving return. While safe ICRL remains an open problem in full generality, our method represents the strongest approach we are aware of.
>
> > The motivation for the proposed method is insufficiently clear. The stated goal of Safe ICRL is to enable more efficient adaptation to OOD scenarios, yet the main technical contribution appears to be online reinforcement pretraining—a technique that is common in online safe RL. The authors should more thoroughly explain why reinforcement pretraining improves OOD adaptation. Is it because the online phase yields more or higher-quality safe trajectories for training? If the benefit is merely due to increased quantity or quality of data, then the claimed motivation rooted in ICRL may be inconsistent.
>
> Reinforcement pretraining here is an umbrella term we use to distinguish the two routes to ICRL (reinforcement vs. supervised distillation), not a standalone contribution borrowed from standard safe RL. Our motivation isn’t "efficiency," but that existing ICRL methods (which are already efficient) ignore safety constraints and break when naively transferred to safe settings. We propose a safe ICRL variant to fix that. The OOD benchmarks are simply a stress test to demonstrate the strong generalization of the learned in-context safe adaptation.
>
> > The experimental environments are relatively simple and too few. The authors should evaluate on more tasks in SafetyGymnasium, e.g., construct multi-task scenarios in MuJoCo Velocity tasks by varying safe velocity thresholds, etc.
>
> we agree that adding a SafetyGymnasium Velocity multi-threshold suite would broaden the empirical picture, and we’ll include this experiment in the final version. We do want to note, though, that the Velocity tasks are considerably easier than our dark-navigation setups: the policy largely boils down to speeding up until the cost activates and then holding pace, whereas SafeDarkMujoco requires memory-driven navigation under partial observability (remembering obstacles, routing around them, and finding the goal), which is a stricter test of safe in-context adaptation.
>
> > The supervised pretraining baseline is purely offline, whereas the proposed method is offline + online; the settings and amounts of training data therefore differ, making the comparison unfair and insufficient to demonstrate the advantage of the proposed method.
>
> The supervised baseline is our best-faith extension of prior ICRL/algorithm-distillation methods to the constrained setting; its offline nature is a defining limitation of that line, not a cherry-picked choice. Our goal is to compare the two viable routes to safe, update-free ICRL, and the results show that online reinforcement pretraining is necessary for safety-aware in-context adaptation. If the reviewer has a specific safe in-context baseline in mind, we’re happy to add it.

---

> ### Author Response · Authors · 2025-11-22
>
> > The paper does not adequately compare EPPO with other online safe RL algorithms (e.g., RCPO [1], CVPO [2], CAL [3], RESPO [4]); only a comparison with a naive primal–dual method in the appendix for a single environment is shown, and cost results are not provided, which is unconvincing.
>
> Those methods are standard update-based online safe RL per environment and are not designed for ICRL, so they’re not the closest comparators to EPPO’s setting. That said, our “naive primal–dual” ablation is essentially the RCPO-style penalty implemented with DDPG. We can add further online safe-RL baselines if the reviewer wants a particular one prioritized.
>
> > The authors should also compare with more recent baselines in the ICRL literature to demonstrate the effectiveness of their cost-oriented design in the safe setting.
>
> We appreciate the reviewer’s concern. To the best of our knowledge, there are no prior ICRL algorithms that address constrained optimization [4]. The supervised pretraining method we include as a "baseline" is in fact an extension of prior ICRL approaches [2, 3] that we developed specifically to serve as a fair comparison point in the constrained setting. If the reviewer is aware of a safe in-context RL baseline we should include, we would be happy to extend the experiments accordingly.
>
> > Additionally, in OOD experiments, comparisons should be made to meta-safe-RL methods under consistent offline/online settings.
>
> We follow prior ICRL work in treating update-free in-context methods as a distinct class from update-based safe meta-RL [1–4].
>
> > Were the reported cost values in the experiments normalized? If yes, what kind of normalization was used? If not, given that the cost values are already small in the zero-shot setting, does this indicate that the experimental environments are too simple in terms of safety constraints?
>
> Costs are not normalized. In these environments, a policy that never moves incurs zero cost, so small zero-shot costs mainly reflect conservative behavior rather than an “easy” safety constraint. What makes a benchmark hard is achieving the task objective while keeping cost low. We show that our safe extension of algorithm distillation (supervised pretraining) fails to do so, and that standard safe RL like RCPO does not reliably improve even on DarkRoom. Together, these results indicate the tasks, especially SafeDarkMujoco, are non-trivial safety benchmarks that only our reinforcement-pretraining approach solves rather reliably.
>
> [1] Lee, Jonathan, et al. "Supervised Pretraining Can Learn In-Context Reinforcement Learning" NeurIPS 2023
>
> [2] Laskin, Michael, et al. "In-context Reinforcement Learning with Algorithm Distillation" ICLR 2023
>
> [3] Dai, Zhenwen, et al. "In-context Exploration-Exploitation for Reinforcement Learning" ICLR 2024
>
> [4] Raparthy, Sharath Chandra, et al. "Generalization to New Sequential Decision Making Tasks with In-Context Learning" ICML 2024

---

> > ### Comment · Reviewer_aqYG · 2025-11-24
> >
> > Thank you very much for the authors’ detailed responses. These replies have indeed addressed some of my concerns, but several concerns still remain.
> >
> > 1. **Contribution** — We acknowledge that the main contribution of this work lies in being the first to study the previously overlooked problem of Safe ICRL, and in demonstrating that reinforcement learning can achieve better performance than supervised pretraining.
> > 2. **Problem setting** — I accept that the dark setting is a standard POMDP used for evaluating ICRL algorithms, and that the proposed method reduces the safety gap compared with previous ICRL approaches. However, all these answers rely on a shared premise: that we are already committed to solving the ICRL problem. My concern is why the problem of Safe ICRL is important in the first place. In real-world scenarios, what situations actually require Safe ICRL? What kinds of constraints cannot be detected by sensors such as radar, and therefore need to be identified through policy exploration?
> > 3. **Comparing pretraining with reinforcement learning** — We still believe that the current comparison is unfair due to the mismatch in dataset size. To more fairly evaluate whether reinforcement learning is more effective than supervised pretraining for Safe ICRL, there are two possible approaches. First, convert reinforcement learning from an online to an offline setup, ensuring that no additional data is used. Second, increase the amount of offline data used for supervised pretraining so that it matches the data available to online reinforcement learning.
> > 4. **More environments** — Presenting results on Velocity tasks—consistent with the DSRL setting—would strengthen the empirical evidence of the paper.
> > 5. **Comparing EPPO with other online safe RL baselines** — We believe that methods like RCPO still fall short of current state-of-the-art online safe RL approaches. Comparing EPPO with methods such as CVPO and RESPO would therefore be more convincing.
> > 6. **Comparison with other baselines** — We acknowledge that there appears to be a lack of established baselines for Safe ICRL or context-based online safe meta-RL.
> >
> > Taking all the responses into account, if additional experimental results are provided, I am willing to update my score to 4.

---

### Official Review · Reviewer_C8L7 · 2025-10-30

**Soundness:** 2
**Presentation:** 2
**Contribution:** 2
**Rating:** 4
**Confidence:** 3

**Summary:**

This paper proposes Safe In-Context Reinforcement Learning (Safe ICRL) by introducing Constrained Markov Decision Processes (CMDPs) into the in-context RL framework.  The approach consists of two parts: Safe Supervised Pretraining and Safe Reinforcement Pretraining, each with its own optimization algorithm designed to handle CMDP constraints.  Experiments are conducted mainly on the SafeDarkRoom and SafeDarkMujoco environments.

**Strengths:**

-   The paper provides an attempt at theoretical analysis of the proposed optimization method.

**Weaknesses:**

1.  The CMDP formulation in the paper appears to consider only a single cost function, which limits generality.

2.  In Theorem 1, the logical order of the proof is somehow  problematic — the existence of the fixed point  should be established before proving that it corresponds to a primal optimal solution.

3.  The experimental evaluation is insufficient. The paper does not compare against established ICRL baselines, making it difficult to assess effectiveness and safety guarantees relative to the state of the art.

4.  The language and structure of the paper need further polishing and clarification.

**Questions:**

1.   For the proposed iterative algorithm:  Does it converge to a fixed point?
2.  Supervised pretraining behavior: In SafeDarkMujoco-Point and SafeDarkMujoco-Car, supervised pretraining does not produce the expected monotonic increase in reward with episode return. What explains this mismatch ？
3.  Cost-to-go vs. return: Figure 2b shows no clear correlation between cost-to-go and return, suggesting that cost-to-go does not effectively control cost tolerance under supervised training. Why does this occur？

---

> ### Author Response · Authors · 2025-11-22
>
> Thank you for your constructive review. We have provided the following response:
>
> > The CMDP formulation in the paper appears to consider only a single cost function, which limits generality.
>
> We would like to clarify that we use the standard CMDP formulation [1-3]. Multi-objective safety problems, where multiple cost functions are handled simultaneously, form a parallel research line that we deliberately keep out of scope. That said, our reinforcement pretraining method is, in principle, extensible to multiple cost functions by introducing additional Lagrange multipliers in the objective. This preserves Theorem 1 with only minor adjustments, although we have not investigated its practical effectiveness.
>
> > In Theorem 1, the logical order of the proof is somehow problematic — the existence of the fixed point should be established before proving that it corresponds to a primal optimal solution.
>
> We thank the reviewer for their comment. We have swapped the two statements in Theorem 1.
>
> > For the proposed iterative algorithm: Does it converge to a fixed point?
>
> Theoretically, a full convergence proof to an exact fixed point is beyond the scope of this work, and we now clarify this in the text.
>
> > The experimental evaluation is insufficient. The paper does not compare against established ICRL baselines, making it difficult to assess effectiveness and safety guarantees relative to the state of the art.
>
> We appreciate the reviewer’s concern. To the best of our knowledge, there are no prior ICRL algorithms that address constrained optimization. The supervised pretraining method we include as a "baseline" is in fact an extension of prior ICRL approaches [4, 5] that we developed specifically to serve as a fair comparison point in the constrained setting. If the reviewer is aware of a safe in-context RL baseline we should include, we would be happy to extend the experiments accordingly.
>
> > Supervised pretraining behavior: In SafeDarkMujoco-Point and SafeDarkMujoco-Car, supervised pretraining does not produce the expected monotonic increase in reward with episode return. What explains this mismatch ？
> > Cost-to-go vs. return: Figure 2b shows no clear correlation between cost-to-go and return, suggesting that cost-to-go does not effectively control cost tolerance under supervised training. Why does this occur？
>
> Both issues reflect the same limitation of supervised distillation in SafeDarkMujoco: the offline MuJoCo logs are too small and don’t span the RTG/CTG trade-off space (especially CTG) well enough, so the model never learns a reliable conditioning. In DarkRoom, a near-covering dataset over phases and cost tolerances is practical because the task is low-dimensional/discrete. This matches with our appendix ablations showing strong sensitivity of supervised pretraining to dataset size.
>
> [1] Tessler, Chen, et al. "Reward Constrained Policy Optimization" ICLR 2019
>
> [2] Yang, Tsung-Yen, et al. "Projection-Based Constrained Policy Optimization" ICLR 2020
>
> [3] McMahan, Jeremy, et al. "Anytime-Constrained Reinforcement Learning" AISTATS 2024
>
> [4] Laskin, Michael, et al. "In-context Reinforcement Learning with Algorithm Distillation" ICLR 2023
>
> [5] Zisman, Ilya, et al., “Emergence of In-Context Reinforcement Learning from Noise Distillation,” ICML 2024

---

### Official Review · Reviewer_GrcY · 2025-11-01

**Soundness:** 4
**Presentation:** 3
**Contribution:** 2
**Rating:** 4
**Confidence:** 3

**Summary:**

This paper introduces and formalizes the problem of Safe In-Context Reinforcement Learning. The goal is for an agent, after a pretraining phase, to adapt to new, out-of-distribution tasks and satisfy safety constraints without any parameter updates. The agent must rely solely on its context to learn and adapt its behavior. Authors propose safe supervised pretraining that imitates learning histories conditioned on return- and cost-to-go, similar to algorithm distillation and decision transformer; and safe reinforcement pretraining that iteratively updates the policy and Lagrangian multipliers in an online manner, called Exact Penalty Policy Optimization (EPPO). Experiments on variants of previous safe/meta-RL benchmarks show that EPPO can achieve good performance while adapting to cost constraints, but safe supervised pretraining struggles in some environments.

**Strengths:**

- Tackles an interesting problem with potential importance in areas like robotics.

- Evaluation is performed in OOD environments, a rather challenging setup.

- Theoretical guarantees are provided.

**Weaknesses:**

- Both methods the authors proposed seem to be rather straightforward extensions of existing algorithms with limited technical novelty.

- Comparison with baselines is lacking. While it is stated that methods in this paper are parameter-update-free, they can arguably still be compared with safe meta-RL approaches with parameter updates under fair setups (e.g. same online interaction budget).

- The authors discussed the issue of the number of evaluation episodes in the methods section but did not conduct any OOD experiments in this regard.

**Questions:**

- How does the methods compare with existing safe meta-RL approaches?

- Can EPPO maintain its performance when given OOD episode number $K$?

---

> ### Author Response · Authors · 2025-11-22
>
> Thank you for your thorough review. We have provided the following response:
>
> > Both methods the authors proposed seem to be rather straightforward extensions of existing algorithms with limited technical novelty.
>
> We appreciate the request for clarity on novelty, but we respectfully disagree.
>
> **Problem side:** We show that imposing safety constraints in the in-context RL setting is not a trivial modification: naive extensions of ICRL with constraint penalties are unstable and fail to learn well-performing policies (Figure 1b, Figure 2b, and Figure 4).
>
> **Solution side:** We analyze the source of this instability and propose a rigorous surrogate objective that uses the constraint structure of ICRL and succeeds on OOD tasks. Although Algorithm 1 differs from a naive DDPG-Lagrangian implementation by only a few lines, this masks substantial design choices. RL algorithms have many interdependent components, and our final method is the result of multiple iterations on alternatives that were either unsupported theoretically or failed to generalize on the OOD benchmark. We also encourage the reviewer to refer to the reinforcement-pretraining README in the supplementary material, which visually depicts our solution.
>
>
> > Comparison with baselines is lacking. While it is stated that methods in this paper are parameter-update-free, they can arguably still be compared with safe meta-RL approaches with parameter updates under fair setups (e.g. same online interaction budget).
> > How does the methods compare with existing safe meta-RL approaches?
>
> We follow prior ICRL work in treating update-free in-context methods as a distinct class from update-based safe meta-RL [1–4]. Matching only the online interaction budget does not make the comparison fair, because update-based methods rely on gradient updates at test time (and thus additional optimization dynamics, hyperparameters, and compute), whereas our setting is explicitly "adapt via context only."
>
> > The authors discussed the issue of the number of evaluation episodes in the methods section but did not conduct any OOD experiments in this regard.
> > Can EPPO maintain its performance when given OOD episode number?
>
> We apologize for the confusion. What we are explaining in lines 223-226 are concerned with the *training* dynamics: in Lagrangian-style training, if we expect to evaluate over $K$ episodes, we should initialize with $K$ (or more) multipliers. If training rollouts contain fewer episodes that the number of multipliers, the later multipliers are updated on a slower timescale, which can destabilize optimization. We’ve revised the wording to make this distinction explicit.
>
> [1] Lee, Jonathan, et al. "Supervised Pretraining Can Learn In-Context Reinforcement Learning" NeurIPS 2023
>
> [2] Laskin, Michael, et al. "In-context Reinforcement Learning with Algorithm Distillation" ICLR 2023
>
> [3] Dai, Zhenwen, et al. "In-context Exploration-Exploitation for Reinforcement Learning" ICLR 2024
>
> [4] Raparthy, Sharath Chandra, et al. "Generalization to New Sequential Decision Making Tasks with In-Context Learning" ICML 2024

---

### Official Review · Reviewer_7od8 · 2025-11-01

**Soundness:** 3
**Presentation:** 2
**Contribution:** 2
**Rating:** 4
**Confidence:** 2

**Summary:**

This paper studies the safety problem in in-context reinforcement learning. The problem is formalized as maximizing the accumulated return from different episodes in context, while ensuring each episode does not violate the cost constraint. The paper proposes EPPO, a safe reinforcement pre-training algorithm that solves the problem and demonstrates its effectiveness through experiments. In particular, EPPO modifies the dual problem of the original optimization problem so that the optimization does not need the knowledge of in-context episodes, while being provably equivalent to optimizing the original problem.

**Strengths:**

1. This paper studies a relevant problem to an emergent training paradigm, a.k.a. the safety problem in ICRL.
2. The proposed algorithm EPPO provably solves the original optimization problem (maximizing reward constrained by fixed cost budget), while being parameter-free in the number of evaluation episodes.

I am not familiar with the literature on ICRL, so I give a low confidence score.

**Weaknesses:**

1. Though the paper mentioned applications of ICRL in language tasks, the safety framework does not seem appropriate for languages. RL in language models is usually not formalized in the way of MDP, and it is usually hard to define a cost function.
2. The approach requires that in the pre-training phase, we already know what the cost function is, whereas this is not realistic in a lot of practical scenarios.
3. The writing is confusing in some paragraphs. For example, I do not understand why the policy in reinforcement pre-training also depends on CTG and RTG.

Minor typo:
1. Line 163: can depend

**Questions:**

1. Is the framework described in the paper applicable to language tasks? If yes, how would you solve the problem of defining the cost function, and that RL in language model is usually not formalized in MDP?
2. Success of pretrained models greatly depends on the usage of offline data, and their general capability to adapt to different downstream tasks. Is it possible to find a way so that we do not need knowledge about the safety constraints during pre-training?
3. Compared to regular training (without considering safety), how much does safety pre-training impact the capabilities of the models?

---

> ### Author Response · Authors · 2025-11-22
>
> Thank you for your insightful review. We have provided the following response:
>
> > Though the paper mentioned applications of ICRL in language tasks ...
>
> The sentence ("For example, by using ICRL at LLMs’ inference time, LLMs can significantly improve on challenging tasks") in the introduction was intended as an example of usefuleness of ICRL.
>
> Also, we think there's a conflation of "training to enable ICRL" with "already possessing ICRL." In-context RL is a capability: The capability of a sequence model to consume interactions with a new MDP to improve on that MDP. Those LLM results are not concerned with the training process that have resulted in such capability. In contrast, we focus on the training algorithms.
>
> > ... the safety framework does not seem appropriate for languages.
> > Is the framework described in the paper applicable to language tasks? If yes, how would you solve the problem of defining the cost function, and that RL in language model is usually not formalized in MDP?
>
> That's an insightful question. The safety framework would be still valid given some cost function with language input. The proposed reinforcement pretraining method works with any black-box cost function. This way, the cost function could simply be another language model or cost models trained from human preference. For the MDP formalization, note that ICRL reqires *multi-turn* interaction with the feedback source. In this setting, there are formalizations [1] that match those in RL.
>
> > The approach requires that in the pre-training phase, we already know what the cost function is, whereas this is not realistic in a lot of practical scenarios.
> > Success of pretrained models greatly depends on the usage of offline data, and their general capability to adapt to different downstream tasks. Is it possible to find a way so that we do not need knowledge about the safety constraints during pre-training?
>
> We follow the standard meta-RL benchmark setup [2] where a reward/cost signal is available during pretraining to define tasks. Importantly, we do not assume knowledge of the test cost: the evaluation CMDP is unseen and different from pretraining tasks, and the state–action regions that incur cost shift between train and test. Learning safe adaptation without any cost access during pretraining is a valuable but orthogonal direction
>
> > I do not understand why the policy in reinforcement pre-training also depends on CTG and RTG.
>
> We're sorry for the confusion. To clarify: our reinforcement pre-training policy does **not** depend on RTG. RTG/CTG inputs are used only in the supervised-pretraining. In the reinforcement-pretraining section we explicitly state the opposite: "policies … [are] in the form of (5) **but without RTG**, i.e., the policy can depend on the context and the CTG." Moreover, at evaluation we "**focus on controlling the CTG exclusively** … [and] pursue the maximal possible return for a given CTG." In short, reinforcement pre-training does not require an RTG signal and the algorithm maximizes return subject to the cost constraint, rather than imitating a specified return.
>
> > Compared to regular training (without considering safety), how much does safety pre-training impact the capabilities of the models?
>
> Safety pre-training trades return for constraint satisfaction. In no-cost sanity checks, convergence was faster and the learned policy was simpler (for example, in gridworld the agent spirals outward from the start, then goes straight to the goal once the goal is found), with returns comparable to our safe-ICRL agent under a very large budget. In Experiment Q(ii), tightening the cost budget monotonically reduces return and Figure 2 quantifies the drop. Thus, with generous budgets the gap is negligible, but as the budget tightens, performance degrades predictably.
>
> [1] Song, Kefan, et al. "Reward Is Enough: LLMs Are In-Context Reinforcement Learners." arxiv:2506.06303
>
> [2] Beck, Jacob, et al. "A tutorial on meta-reinforcement learning." Foundations and Trends in Machine Learning 2025

---

### Meta-Review · Area_Chair_ro1q · 2026-01-02

**Summary:**

On the positive side, the reviewers had questions regarding the use of constraints (expressed as a cost function) in the context of language tasks and the use of a single cost function, which were well addressed in the rebuttal.
On the negative side, the reviewers had concerns regarding the novelty of the proposed approach and wondered if previous safe RL methods could be used for the same approach. Furthermore, the empirical evaluation was modest and could benefit from further baselines and environments. Additionally, the reviewers agree that the writing quality should be improved and the paper needs a revision to improve clarity. Finally, the motivation for the problem formulation could be stronger.

**Reviewer Concerns:**

- `7od8` noticed that the approach requires knowledge of the cost function, pointed out the low writing quality, and questioned the applicability of the framework to language tasks. The issues with the writing quality remain open. Indeed, the empirical evaluation focuses on the traditional constrained RL tasks; nevertheless, the rebuttal provided support that the formulation can be used for safety in language tasks, although not in the traditional way as pointed out by the reviewer, which also demonstrates that the approach only needs black box access to the cost function which is less restrictive than explicit knowledge of the cost function.
- `GrcY` mentioned the approach has limited novelty (but did not support this claim),  mentioned the need for empirical comparison with safe meta RL, and questioned details about the empirical evaluation in OOD regime. In general, the authors provided a strong rebuttal demonstrating that the empirical comparison with safe meta RL is unfair. Furthermore, the rebuttal was sufficient to clarify the questions of the reviewer.
- `C8L7` questioned the formulation with a single cost function, the order in which the theoretical results were presented, and the lack of state-of-the-art baselines from ICRL, and noticed that the writing quality is low.  The rebuttal was sufficient to clarify the questions from the reviewer, but did not address the issues regarding the writing quality.
- `aqYG` questioned the limited number of experiments in the empirical evaluation with a small number of baselines, and found the motivation weak and unclear.

**Reviewer Scores:**

- `7od8`: 4 -> 6
- `GrcY`: 4 -> 6
- `C8L7`: 4 -> 4
- `aqYG`: 2 -> 4

---

### Decision · Program_Chairs · 2026-01-26

Reject